# NextLocMoE: Enhancing Next Location Prediction via Location-Semantics Mixture-of-Experts and Personalized Mixture-of-Experts

## Abstract

Next location prediction is a key task in human mobility modeling. Existing methods face two challenges: (1) they fail to capture the multi-faceted semantics of real-world locations; and (2) they struggle to model diverse behavioral patterns across user groups. To address these issues, we propose NextLocMoE, a large language model (LLM)-based framework for next location prediction, which integrates a dual-level Mixture-of-Experts (MoE) architecture. It comprises two complementary modules: a Location Semantics MoE at the embedding level to model multi-functional location semantics, and a Personalized MoE within LLM's Transformer layers to adaptively capture user behavior patterns. To enhance routing stability and reliability, we introduce a historical-aware router that integrates long-term historical trajectories into expert selection. Experiments on multiple real-world datasets demonstrate that NextLocMoE outperforms existing methods in accuracy, transferability, and interpretability. Code is available at: https://anonymous.4open.science/r/NextLocMOE-BAC8.

## 1 Introduction

Predicting a user's next location from past trajectory has become a critical task in domains like intelligent transportation (Liu et al., 2020), personalized service (Li et al., 2024b), and urban management (Yang et al., 2024b). The goal is to model user mobility patterns and moving intentions to infer the most likely next destination. Early approaches relied on recurrent neural networks (Chung et al., 2014; Graves, 2012) to capture temporal dependencies. With the emergence of Transformer (Vaswani et al., 2017), methods like MHSA (Hong et al., 2023b), CLLP (Zhou et al., 2024), and GETNext (Yang et al., 2022) were developed to capture complex spatiotemporal interactions. Recently, large language models have been applied to this task. Llama-Mob (Tang et al., 2024), LLMMob (Wang et al., 2023b), and SILO Sun et al. (2025) leverage LLMs' language understanding, reasoning ability, and pre-trained world knowledge to enhance predictive performance.

While existing methods have made notable progress, they still face two major challenges. First, most models learn a single embedding for each location, which may not fully capture the multi-functional semantics of real-world locations. For example, a location in city center may simultaneously serve commercial, residential, and educational purposes. Compressing such diverse signals into a single embedding can lead to semantic compression and even embedding collapse—a phenomenon where the embedding space becomes low-rank and loses diversity (Guo et al., 2023). This limits representation richness and weakens downstream prediction. Second, most methods adopt a shared set of parameters for all users. Though this design captures user diversity to some extent, it lacks structural mechanisms to disentangle heterogeneous mobility patterns. This often leads to entangled representations that blend signals across user groups (e.g., students, office workers, tourists), making it difficult to specialize or interpret distinct mobility behaviors. Recent studies echo this concern: Su et al. (2023) argue that a single shared model overlooks key behavioral differences, and Zhang et al. (2025) show that shared Transformers act as low-pass filters, suppressing informative high-frequency signals. Some models try to introduce personalized modeling via user embeddings (Zhou et al., 2024; Yang et al., 2022), but they face two shortcomings: (1) reliance on user IDs, which poses challenges in cold-start scenarios with unseen users, and (2) limited interpretability, as user embeddings provide little insight into learned behavioral patterns. To tackle these challenges, we

propose NextLocMoE, a dual-level Mixture-of-Experts (MoE) based LLM framework for next location prediction, which jointly models location semantics and user behavioral patterns.

To model location semantics, we design Location Semantics MoE, which enriches location representations by combining a shared spatial embedding with expert embeddings specialized for different functional roles. The shared embedding encodes geographic coordinates to capture general spatial features. The router of this MoE module activates the top-$k$ most relevant location function-specific experts, each encoding the same coordinates into a function-aware embedding. This results in multiple expert embeddings that reflect the diverse semantics a single location may exhibit. To inject semantic priors and improve interpretability, each expert is initialized with LLM-encoded natural language descriptions of predefined location function categories.

To capture user behavioral patterns, NextLocMoE integrates Personalized MoE into selected Transformer layers of the LLM backbone by replacing the feedforward networks (FFNs). This design enables group-level personalization while preserving LLM's semantic encoding capacity. We predefine a set of user groups and encode their natural language descriptions using LLM to obtain group-specific embeddings. The router then combines these embeddings with user's historical trajectory representation to dynamically select the most relevant expert submodules. Unlike the top-$k$ routing strategy used in Location Semantics MoE, Personalized MoE employs a confidence threshold based expert activation mechanism inspired by (Huang et al., 2024). This design is motivated by two considerations: (1) users may exhibit varying degrees of behavioral ambiguity, making it preferable to flexibly adjust the number of active experts; and (2) limiting expert activation reduces computational overhead. As a result, Personalized MoE activates fewer experts for users with consistent behavioral patterns, while allocating more capacity to users with uncertain or mixed behaviors.

To improve long-term behavior awareness and expert selection stability in both MoE modules, NextLocMoE introduces a historical-aware router that explicitly incorporates historical trajectories into expert routing. In conclusion, our main contributions are summarized as follows:

- We propose NextLocMoE, a novel LLM-based framework that integrates Mixture-of-Experts (MoE) into next location prediction. It comprises (i) a Location Semantics MoE for modeling the multi-functional roles of locations, and (ii) a Personalized MoE to capture diverse user behavioral patterns. Each module is guided by expert-specific priors and customized routing strategy.

- We introduce a historical-aware router that incorporates long-term historical trajectory into expert selection, enhancing the contextual stability and reliability of expert routing.

- Extensive experiments on multiple real-world datasets demonstrate that NextLocMoE consistently outperforms other baselines under both fully-supervised and zero-shot settings. Case studies further highlight the model's ability to provide interpretable predictions.

## 2 RELATED WORK

### 2.1 NEXT LOCATION PREDICTION

Next location prediction aims to forecast the most probable location a user will visit, based on past trajectories. Early methods relied on recurrent neural networks like GRU (Chung et al., 2014) and LSTM (Graves, 2012), to capture temporal dependencies. DeepMove (Feng et al., 2018) enhances trajectory representation by jointly modeling short-term interests and long-term preferences. However, these methods struggle with long-range dependencies and suffer from limited parallelism, which constrains their scalability. With the rise of Transformer (Vaswani et al., 2017), attention-based methods have become the mainstream in next location prediction. MHSA (Hong et al., 2023b) models transition relationships between locations via multi-head self-attention. CLLP (Zhou et al., 2024) integrates local and global spatiotemporal contexts to better capture dynamic user interests. GETNext (Yang et al., 2022) introduce global trajectory flow graphs and graph-enhanced Transformer models, leveraging collaborative mobility signals to improve predictive performance.

In recent years, breakthroughs in large language models (Achiam et al., 2023; Liu et al., 2024a; Touvron et al., 2023) have inspired researchers to explore their potential in next location prediction. Llama-Mob (Tang et al., 2024) and LLMMob (Wang et al., 2023b) design task-specific prompts, while NextLocLLM (Liu et al., 2024c) leverages LLM as both a semantic enhancer and a predic-

tor. These methods exploit pre-trained world knowledge and reasoning capabilities to improve the semantic understanding of user mobility and enhance both prediction accuracy and generalization.

Despite these advances, two key limitations remain. First, most models assign a single embedding to each location, which fails to capture the multifaceted semantics of real-world locations. Second, most models use a shared set of parameters for all users, overlooking behavioral differences among user groups. These limitations constrain both the accuracy and the adaptability of existing methods in real-world settings. Therefore, we propose NextLocMoE, a novel framework that introduces a dual-level Mixture-of-Experts architecture to model both location semantics and user behaviors. For a more detailed discussion of related work on next location prediction, please refer to App. A.1.

### 2.2 MIXTURE OF EXPERTS

Mixture of Experts (MoE) is designed to enhance model expressiveness and computational efficiency. It maintains a pool of expert networks and dynamically activates a subset of them for each input, allowing MoE-based models to match the performance of larger architectures while keeping computation cost low. MoE has achieved notable success in natural language processing, with prominent examples like GShard (Lepikhin et al., 2020), Switch Transformer (Fedus et al., 2022), and DeepSeekMoE (Dai et al., 2024). It has also been explored in sequence modeling tasks, as demonstrated by Time-MoE (Shi et al., 2024), Moirai-MoE (Liu et al., 2024d), and Graph MoE (Huang et al., 2025). However, MoE is still underexplored in next location prediction. To bridge this gap, we introduce NextLocMoE, which incorporates dual-level MoE modules targeting location semantics and user behavioral patterns, paving the way for MoE architecture in next location prediction. For a more detailed discussion of MoE related work, please refer to App. A.2.

## 3 PROBLEM FORMULATION

Let $\mathcal{L} = \{loc_1, loc_2, \ldots, loc_n\}$ be the set of locations, where each $loc$ is a triplet $(id, x, y)$, with $id$ being the location identifier and $(x, y)$ its spatial coordinates. We define the temporal context set as $\mathcal{T} = \{(w, d) \mid w \in [0, 6], \ d \in [0, 23]\}$, where $w$ denotes day-of-week and $d$ denotes time-of-day. Let $\mathcal{D}_{\nabla} = \{dur\}$ be the set of stay durations, indicating how long a user stays at a given location.

*Definition 1 (**Record**)* A record is defined as a tuple $s = (loc, (w, d), dur) \in \mathcal{L} \times \mathcal{T} \times \mathcal{D}_{\nabla}$, which indicates that a user visited location $loc$ for $dur$ hours at hour $d$ on day-of-week $w$.

*Definition 2 (**Historical and Current Trajectory**)* A user's mobility sequence can be partitioned into two disjoint segments: historical trajectory and current trajectory. The former is denoted as $S_h = \{s_{t_1}, s_{t_1+1}, \ldots, s_{t_1+M-1}\}$, which contains $M$ records used to model the user's long-term behavioral preferences. The latter is denoted as $S_c = \{s_{t_2}, s_{t_2+1}, \ldots, s_{t_2+N-1}\}, (t_2 \geq t_1 + M)$, which includes the most recent $N$ records and is used to capture the user's short-term intent. Typically, $M > N$, ensuring that the historical trajectory spans a longer behavioral window.

*Definition 3 (**Next Location Prediction**)* Given a user's historical trajectory $S_h$ and current trajectory $S_c$, next location prediction aims to infer the identifier $id$ of the most likely next location $loc_{t_2+N}$.

## 4 METHODOLOGY

### 4.1 OVERALL ARCHITECTURE

Fig. 1 depicts the overall architecture of NextLocMoE. It takes user's historical and current trajectory as input. Each record is mapped into spatial-temporal embedding (Sec. 4.2) by encoding spatial coordinates $(x, y)$, temporal context ($w$ and $d$), and stay duration $dur$, which are then concatenated. For current trajectory, we employ Location Semantics MoE (Sec. 4.3) to enrich spatial embedding with location function semantics. The function-aware spatial embedding is then combined with temporal embeddings to form the enhanced current trajectory embedding.

Next, we concatenate historical trajectory embedding, enhanced current trajectory embedding, and a task-specific prompt (See App. D) to construct the full input embedding for LLM backbone (Sec. 4.6). Inspired by (Huang et al., 2024), NextLocMoE employs only the first $L_1 + L_2$ layers of LLM: $L_1$ standard LLM layers and subsequent $L_2$ layers augmented with Personalized MoE

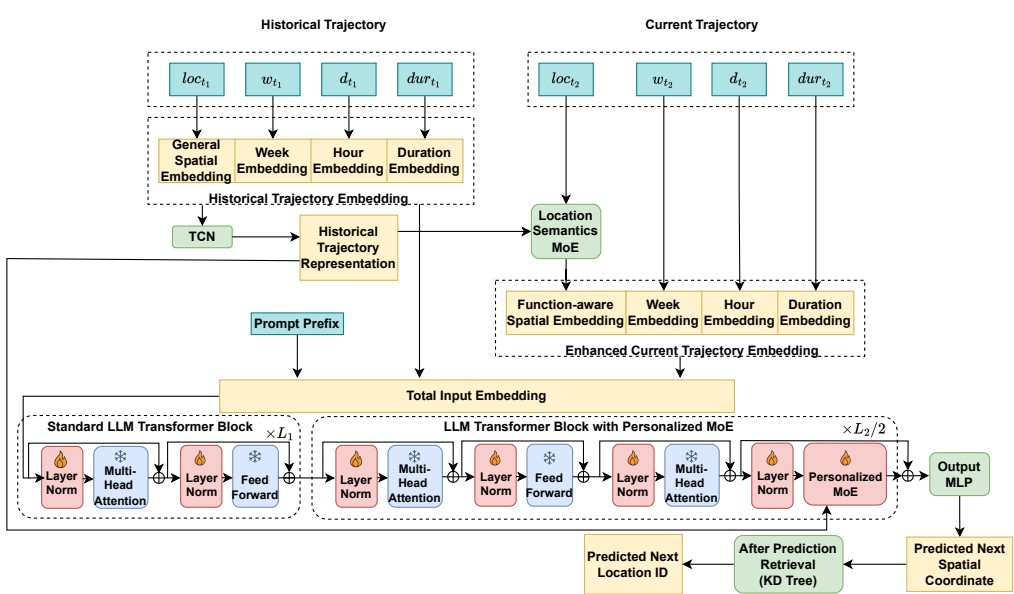

Figure 1: Overall architecture of NextLocMoE, a Mixture-of-Experts enhanced LLM framework for next location prediction. It introduces a Location Semantics MoE to capture multi-functional spatial semantics (see Fig. 2(a), a Personalized MoE to model behavioral differences across user groups (see Fig. 2(a)), and a historical-aware router that incorporates long-term trajectory into expert routing.

(Sec. 4.4) to model user behavioral patterns. To improve expert routing robustness and reliability, NextLocMoE introduces a historical-aware router (Sec. 4.5) that incorporates long-term historical trajectories into expert selection. To reduce parameter overhead, we fine-tune only the FFN sub-networks within MoE experts and all LayerNorm layers, freezing the remaining backbone layers.

The final output of NextLocMoE is the predicted spatial coordinate of next location, obtained via an output MLP head. During inference, a post-prediction retrieval module (Sec. 4.7) maps these coordinates to the nearest discrete location ID . This output design is motivated by two considerations: (1) predicting continuous coordinates enables city-agnostic modeling and supports cross-city generalization, whereas direct classification over location IDs is city-specific and does not transfer; (2) the post-prediction retrieval module ensures fair comparison with prior ID-based baselines while adding negligible inference overhead and no intervention on predictive accuracy.

## 4.2 Spatial-Temporal Embedding

In NextLocMoE, each component of a record is embedded through linear projection or embedding lookup. Specifically, spatial coordinates $(x, y)$ and stay duration $dur$ are normalized and projected via linear layers to produce general spatial embedding $\mathbf{e}_{xy}$ and duration embedding $\mathbf{e}_{dur}$. For temporal context, $w$ and $d$ are encoded via lookup tables, yielding temporal embeddings $\mathbf{e}_w$ and $\mathbf{e}_d$.

For $S_h$, we concatenate the above four vectors along feature dimension to obtain historical trajectory embedding $\mathbf{z}_h$, which is used in two ways: as input to the LLM backbone and, after TCN encoding, as input to expert routers of both Location Semantics MoE and Personalized MoE. For each record in $S_c$, we adopt the same procedure to generate initial embedding $\mathbf{e}_c^{(0)}$. This embedding is used as input to Location Semantics MoE, where it is combined with TCN-encoded historical trajectory representation to guide expert selection and generate function-aware spatial embedding. To

## 4.3 Location Semantics MoE

In urban settings, a single location often serves multiple functions (e.g., shopping malls, schools, public services). Encoding such locations with a single vector limits model expressiveness. To

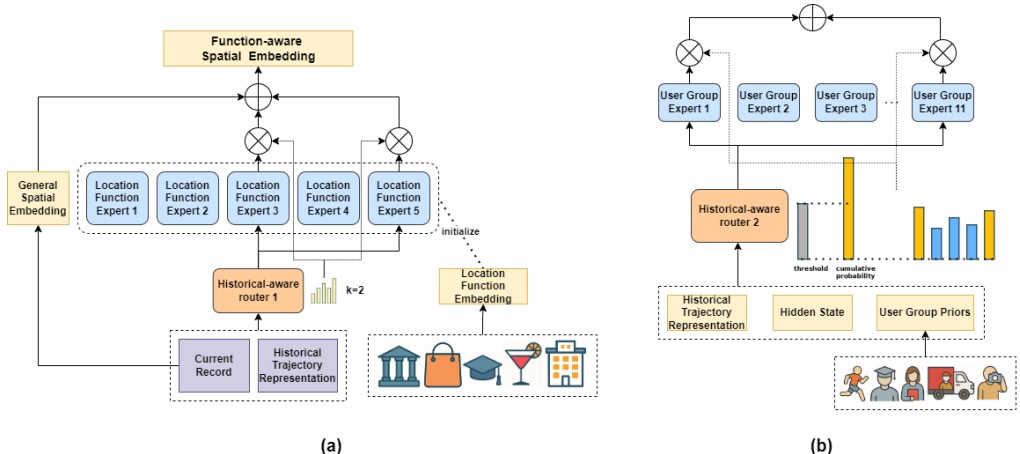

Figure 2: The two expert modules. (a) Location Semantics MoE, (b) Personalized MoE.

address this, NextLocMoE introduces Location Semantics MoE into current trajectory encoding (Fig. 2(a)), enabling fine-grained, function-aware location representations.

This module takes as input the historical trajectory representation $\mathbf{h}^{\text{hist}}$ and the initial embedding of each record in the current trajectory, $\mathbf{e}_c^{(0)}$. The former is obtained by encoding historical trajectory embedding $\mathbf{z}_h$ using a Temporal Convolutional Network (TCN) (Lea et al., 2017):

$$\mathbf{h}^{\text{hist}} = \text{TCN}(\mathbf{z}_h). \tag{1}$$

$\mathbf{e}_c^{(0)}$ and $\mathbf{h}^{\text{hist}}$ are fed into expert router to generate a scoring vector over $K_f$ function experts:

$$\mathbf{r}^{\text{func}} = \text{MLP}([\mathbf{e}_c^{(0)}; \mathbf{h}^{\text{hist}}]) \in \mathbb{R}^{K_f}. \tag{2}$$

$\mathbf{r}^{\text{func}}$ is normalized via softmax to obtain expert selection probabilities $\mathbf{p}_i^{\text{func}}$. The router then selects top-$k$ experts with highest probabilities to enhance the semantic representation of the current record.

Each function expert $\mathbf{f}_i(\cdot)$ is a linear projection that maps spatial coordinates $(x, y)$ to a function-specific embedding. Its structure is identical to the mapping used for general spatial embedding $\mathbf{e}_{xy}$. To promote interpretability and specialization, we predefine a set of location function categories (see App. B) and encode their natural language descriptions using LLM. The resulting LLM-encoded embeddings are used to initialize the parameters of experts, and these experts are then fine-tuned. This semantic initialization serves as a soft prior, introducing meaningful inductive biases that encourage experts to specialize toward distinct functional roles. Previous studies (Kang et al., 2025; Min et al., 2025) demonstrate that such initialization stabilizes optimization, and guides experts to remain aligned with intended semantic roles rather than collapsing into undifferentiated roles throughout training. Consequently, this initialization design ensures that function experts in Location Semantics MoE remain interpretable, facilitating semantic disentanglement and fast convergence.

Given the selected top-$k$ function experts and their routing probabilities $\mathbf{p}_i^{\text{func}}$, we compute the summed location function specialized expert embedding as:

$$\mathbf{e}_{xy}^{\text{func}} = \sum_{i \in \text{top}k(\mathbf{p}^{\text{func}})} \mathbf{p}_i^{\text{func}} \cdot \mathbf{f}_i(x, y). \tag{3}$$

Motivated by Deepseek-MoE(Dai et al., 2024), we treat general spatial embedding $\mathbf{e}_{xy}$ as a shared expert and combine it with $\mathbf{e}_{xy}^{\text{func}}$ to obtain the function-aware spatial embedding $\mathbf{e}_{xy}^{\text{enhanced}}$:

$$\mathbf{e}_{xy}^{\text{enhanced}} = \mathbf{e}_{xy} + \mathbf{e}_{xy}^{\text{func}}. \tag{4}$$

It is worth noting that Location Semantics MoE is applied only to current trajectory records, not to the historical ones. This is based on several considerations: (1) historical trajectories are used to model long-term behavioral patterns, where temporal dynamics outweigh fine-grained semantics; (2) applying MoE to all records incurs high computational and memory costs; (3) function disambiguation is more important for current locations, whose semantics are directly tied to prediction.

## 4.4 PERSONALIZED MoE

To capture behavioral variations across user groups, NextLocMoE integrate Personalized MoE into the upper layers of LLM backbone (Fig. 2(b)). We predefine $K_p$ prototypical user groups (see App. C), each linked to an expert module. For each group, its natural language description is encoded by LLM and transformed into a user group prior $\mathbf{e}_i^{\text{user}}$ $(i = 1, \cdots, K_p)$ through a mean-pooling layer and a linear transformation. These priors provide semantic guidance and distinguish experts by behavioral identity. Although explicit user group labels are not involved, the router leverages historical trajectories and LLM-encoded user group descriptions to dynamically activate relevant experts, enabling the model to capture heterogeneous behavioral patterns without supervision.

Personalized MoE receives the hidden state $\mathbf{x}$ from previous LLM layer, along with historical trajectory representation $\mathbf{h}^{\text{hist}}$. For each expert $i$, it concatenates these with its user group prior:

$$\mathbf{z}_i^{\text{user}} = [\mathbf{x}; \ \mathbf{h}^{\text{hist}}; \ \mathbf{e}_i^{\text{user}}]. \tag{5}$$

$\mathbf{z}_i^{\text{user}}$ is first transformed by a multi-layer perceptron $\text{Fusion}(\cdot)$, and then projected by a linear gating function $\text{Gate}(\cdot)$ to compute the relevance score $\mathbf{r}_i^{\text{user}}$:

$$\mathbf{r}_i^{\text{user}} = \text{Gate}(\text{Fusion}(\mathbf{z}_i^{\text{user}})). \tag{6}$$

Stacking the scores across all experts yields the complete relevance vector:

$$\mathbf{r}^{\text{user}} = \{\mathbf{r}_1^{\text{user}}, \mathbf{r}_2^{\text{user}}, \cdots, \mathbf{r}_{K_p}^{\text{user}}\} \in \mathbb{R}^{K_p}. \tag{7}$$

We apply softmax over $\mathbf{r}^{\text{user}}$ to obtain the selection probability for each user group expert, $p_i^{\text{user}}$.

Unlike top-$k$ routing used in Location Semantics MoE, Personalized MoE adopts a confidence threshold-based expert routing strategy (Huang et al., 2024). We sort experts by their selection probabilities $p_i^{\text{user}}$ and activate them until the cumulative probability exceeds threshold $\tau$:

$$\mathcal{E} = \{i_1, i_2, \ldots, i_m\}, \quad \text{where} \quad \sum_{k=1}^{m} p_{i_k}^{\text{user}} \geq \tau. \tag{8}$$

This allows adaptive expert activation: users with stable mobility patterns activate fewer experts, while those with diverse or ambiguous behaviors activate more. Activated experts perform feedforward transformations on hidden state $\mathbf{x}$ and their outputs $\mathbf{h}_i$ are aggregated via weighted sum:

$$\mathbf{h}^{\text{out}} = \sum_{i \in \mathcal{E}} p_i^{\text{user}} \cdot \mathbf{h}_i. \tag{9}$$

## 4.5 HISTORICAL-AWARE ROUTER

Standard MoEs typically relies solely on current input for expert selection (Fedus et al., 2022), However, users with similar short-term routines may diverge in destination due to long-term behavior differences. For instance, after the same morning routine from home to a metro station, a student may go to university, while an office worker may head to a business district. Ignoring such historical context in expert routing would compromise both semantic and personalized behavior modeling.

To address this, NextLocMoE introduces historical-aware router, which incorporates historical trajectories into expert selection. Specifically, we employ a TCN (Lea et al., 2017) to encode historical embeddings $\mathbf{z}_h$, yielding historical trajectory representation $\mathbf{h}^{\text{hist}}$, which is subsequently integrated into expert routing for both MoE modules. We choose TCN for its ability to efficiently capture long-range temporal dependencies and enable strong parallelization. By incorporating historical trajectory representation, historical-aware router mitigates the over-reliance on local context, stabilizes expert selection, and ultimately improves predictive accuracy and generalization.

## 4.6 STREAMLINED LLM BACKBONE AND EFFICIENT EXPERT ADAPTATION

To reduce computational overhead while maintaining predictive accuracy, NextLocMoE adopts a streamlined design. It retains only the first $L_1 + L_2$ layers of LLM. The lower $L_1$ layers remain LLM layers, while the upper $L_2$ layers replace their original feedforward networks (FFNs) with Personalized MoE. This design is inspired by (Skean et al., 2025), which shows that intermediate LLM

representations are more stable and transferable than top-layer outputs, effectively filtering out high-entropy noise in downstream tasks. By truncating the model at intermediate layers, NextLocMoE preserves semantic encoding capacity while reducing architectural complexity.

To further limit trainable parameters and avoid overfitting, we freeze all multi-head attention modules and non-MoE FFNs in LLM backbone, keeping only LayerNorm layers and Personalized MoE experts trainable. Each user group expert is initialized from the FFN it replaces, ensuring representational continuity. To enhance training efficiency, we apply Low-Rank Adaptation (LoRA) to each user group expert. LoRA introduces a small set of trainable parameters in low-rank subspaces, allowing expert specialization and efficient personalization at minimal computational cost.

### 4.7 TRAINING OBJECTIVE AND POST-PREDICTION RETRIEVAL

The training objective of NextLocMoE combines a regression loss and an expert entropy regularization term. Given a batch of $B$ samples, NextLocMoE predicts the spatial coordinates $(\hat{x}, \hat{y})$ of next location. The regression loss is defined as the mean Euclidean distance to the ground truth $(x, y)$:

$$\mathcal{L}_{\text{dist}} = \frac{1}{B} \sum_{i=1}^{B} \|(\hat{x}_i, \hat{y}_i) - (x_i, y_i)\|_2 .$$ (10)

To encourage confident expert routing in Personalized MoE and reduce unnecessary expert activation, we introduce an entropy regularization term:

$$\mathcal{L}_{\text{entropy}} = -\mathbb{E}_i \sum_j p_{i,j}^{\text{user}} \log p_{i,j}^{\text{user}}.$$ (11)

The final training objective is a weighted combination of the two:

$$\mathcal{L}_{\text{total}} = \mathcal{L}_{\text{dist}} + \lambda \times \mathcal{L}_{\text{entropy}},$$ (12)

where $\lambda$ balances the influence of the entropy regularization term. Unlike some MoE frameworks that impose explicit load-balancing losses (Dai et al., 2024; Huang et al., 2024), we avoid such regularization. Imbalance in expert utilization naturally reflects the heterogeneous distribution of location functions and user behaviors in urban data. Enforcing uniform expert usage would suppress meaningful specialization and force rare but semantically important experts to be underutilized.

During inference, NextLocMoE maps predicted continuous coordinates to discrete location IDs via a KD-Tree nearest neighbor search. This KD-Tree is constructed from candidate location coordinates of target city, and predicted coordinates are queried to retrieve the IDs of the top-k nearest candidates. This mapping serves only as a post-processing step and does not depend on the current location, ensuring that NextLocMoE remains free to predict both nearby and distant transitions.

## 5 EXPERIMENT

To evaluate the effectiveness of NextLocMoE, we conduct comprehensive experiments across several key dimensions: prediction accuracy, cross-city transferability, interpretability, and broader analyses on robustness and model design.

### 5.1 EXPERIMENTAL SETUP

We evaluate NextLocMoE on three human mobility datasets: Kumamoto, Shanghai, Singapore (details in App. F). User-level dataset partitioning (Sun et al., 2021) splits users into training, validation, and test sets in a 7:1:2 ratio. Following (Luo et al., 2021) and (Feng et al., 2018), we adopt Hit@1/5/10 for evaluation. Historical and current trajectory lengths are set to $M = 40$ and $N = 5$. LLM backbone is LLaMA-3.2-3B, with $L_1 = 8$ and $L_2 = 4$. NextLocMoE is trained using Adam optimizer with ReduceLROnPlateau scheduler, on four 32GB Tesla V100 GPUs.

We compare NextLocMoE with a wide range of baselines, including RNN-based models (GRU, LSTM, DeepMove); Transformer-based methods (MHSA, CLLP, LoTNext,GETNext, SEAGET, ROTAN), and LLM-based approaches (LLM4POI, NextLocLLM,Mobility-LLM,AgentMove, SILO,Llama-Mob, LLMMob, ZSNL). Details are available in App. G

Table 1: Fully-supervised next location prediction results.

| Method | Kumamoto | | | Shanghai | | | Sinapore | | |
|---|---|---|---|---|---|---|---|---|---|
| | Hit@1 | Hit@5 | Hit@10 | Hit@1 | Hit@5 | Hit@10 | Hit@1 | Hit@5 | Hit@10 |
| GRU | 3.213% | 6.720% | 8.735% | 19.69% | 25.90% | 29.04% | 2.682% | 6.051% | 7.784% |
| LSTM | 3.192% | 6.483% | 8.514% | 22.03% | 28.81% | 31.33% | 3.197% | 8.698% | 10.46% |
| MHSA | 2.982% | 9.203% | 11.77% | 48.40% | 56.62% | 62.21% | 4.874% | 13.54% | 19.38% |
| DeepMove | 11.11% | 20.71% | 24.46% | 53.48% | 62.13% | 67.70% | 6.650% | 20.00% | 31.08% |
| GetNext | 12.68% | 24.57% | 29.80% | 55.18% | 64.17% | 71.17% | 6.498% | 25.80% | 32.04% |
| CLLP | 10.69% | 17.79% | 21.96% | 56.24% | 65.39% | 72.08% | 7.712% | 26.98% | 34.99% |
| SEAGET | 12.79% | 24.66% | 29.99% | 55.39% | 65.12% | 70.93% | 6.512% | 25.94% | 32.56% |
| NextLocLLM | 13.57% | 24.78% | 31.16% | 59.62% | 66.93% | 72.81% | 7.823% | 30.64% | 36.15% |
| ROTAN | 13.01% | 26.19% | 32.37% | 57.92% | 66.83% | 72.06% | 6.892% | 27.71% | 35.56% |
| LoTNext | 13.58% | 24.96% | 31.22% | 56.48% | 66.56% | 72.59% | 7.398% | 26.19% | 33.46% |
| Mobility-LLM | 13.55% | 24.44% | 31.69% | 56.06% | 64.04% | 73.06% | 7.376% | 25.67% | 32.87% |
| AgentMove | 13.12% | 22.87% | 30.63% | 55.62% | 62.47% | 72.00% | 6.939% | 24.10% | 33.93% |
| SILO | 15.63% | 33.41% | 45.59% | 61.44% | 67.71% | 73.06% | 8.692% | 32.59% | **42.63%** |
| LLM4POI | 13.17% | 26.88% | 30.11% | 58.83% | 67.72% | 72.47% | 7.952% | 31.69% | 38.88% |
| SoloPath | 13.75% | 27.80% | 34.61% | 60.21% | 67.92% | 69.24% | 8.102% | 30.00% | 37.51% |
| Llama-Mob | 15.78% | 33.55% | 43.42% | 61.81% | 69.36% | 73.45% | 8.577% | 32.17% | 41.21% |
| LLMMob | 10.95% | 25.54% | 35.77% | 51.17% | 60.93% | 63.31% | 6.933% | 21.07% | 30.70% |
| ZS-NL | 8.811% | 22.97% | 31.76% | 39.92% | 47.71% | 50.98% | 4.199% | 14.68% | 20.11% |
| NextLocMoE | **17.77%** | **39.19%** | **50.28%** | **64.93%** | **75.88%** | **77.43%** | **9.733%** | **34.34%** | 40.71% |

Table 2: Zero-shot Prediction Result (Kumamoto).

| Method | Hit@1 | Hit@5 | Hit@10 |
|---|---|---|---|
| LLMMob | 10.95% | 25.54% | 35.77% |
| ZS-NL | 8.811% | 22.97% | 31.76% |
| Llama-Mob(Shanghai→) | 15.78% | 33.55% | 43.42% |
| Llama-Mob(Singapore→) | 14.96% | 31.27% | 40.32% |
| NextlocLLM(Shanghai→) | 13.14% | 28.68% | 39.26% |
| NextlocLLM(Singapore→) | 11.73% | 26.95% | 37.53% |
| NextLocMoE(Shanghai→) | 16.02% | 36.06% | 48.42% |
| NextLocMoE(Singapore→) | 15.81% | 34.66% | 47.41% |

Table 3: Inference Time (Kumamoto).

| Method | Time (s) |
|---|---|
| Llama-Mob | 158688 |
| LLMMob | 33408 |
| NextLocMoE | 268 |

## 5.2 EXPERIMENTAL RESULT

We present key results here, while additional findings are in the Appendix, including ablation studies (App J), robustness of post-prediction retrieval (App K), user group-expert activation consistency (App L), routing strategy evaluation (App M), historical trajectory modeling evaluation (App N), LLM backbone comparison (App O), hyperparameter sensitivity (App I) and personalized expert activation (App Q. They further validate the effectiveness, robustness, and generality.

### 5.2.1 FULLY-SUPERVISED PREDICTION COMPARISON

Table. 1 presents the fully-supervised next location prediction results on all three datasets. RNN-based models perform poorly, indicating that local temporal modeling is insufficient to capture the complex spatiotemporal mobility patterns. Methods like CLLP, GETNext, SEAGET, ROTAN, and LLM4POI rely on user IDs or user embeddings, which fail to generalize under user-level partitioning where test users are unseen during training. Llama-Mob, the winner of 2024 HuMob Challenge, performs better but remains limited in modeling multi-functional location semantics and user behavioral patterns. In contrast, NextLocMoE introduces two innovations: Location Semantics MoE for fine-grained semantic modeling of locations, and Personalized MoE for user behavioral patterns. These designs lead to consistent state-of-the-art performance across all datasets and metrics.

### 5.2.2 ZERO-SHOT PREDICTION COMPARISON

To evaluate cross-city generalization, we conduct zero-shot experiments on Kumamoto dataset, where models are evaluated after trained on other cities without any fine-tuning. Since location IDs differ across cities, ID-based non-LLM models cannot be transferred in this setting. Thus, we

compare only transferable methods: Llama-Mob, LLMMob, ZS-NL, NextLocLLM, and our proposed NextLocMoE. As shown in Table 2, NextLocMoE achieves the best across all metrics. We attribute this to its explicit modelling of location semantics and user behavior: the Location Semantics MoE encodes functional semantics agnostic to location IDs, while Personalized MoE adapts to user behavior through role-based experts—enabling robust transfer to unseen cities.

### 5.2.3 INFERENCE TIME

We report the total inference time of the transferable LLM-based models on Kumamoto test set in Table 3. Llama-Mob relies on locally deployed LLM with separate prompt construction per trajectory, resulting in highly sequential and time-consuming inference. LLMMob, offloading computation via external APIs, still suffers from serialized generation. NextLocMoE adopts a unified architecture that supports batch inference and GPU parallelism. It completes inference in 268 seconds—a 600× speedup over Llama-Mob and 120× faster than LLMMob.

### 5.2.4 CASE STUDY

We analyze two representative trajectories from Singapore dataset (Fig. 3). Though their current trajectory exhibit similar spatial patterns, their historical trajectories differ: the first is centered around academic zones, while the second frequently appears in commercial and tourist areas. In Location Semantics MoE, the first case assigns higher weights to Education and Entertainment, while the second favors Entertainment and Commercial. In Personalized MoE, the first user is routed to student and teacher experts, whereas the second strongly activates the tourist expert. These expert assignments align with the corresponding demographic attributes (the first user being a student and the second being a tourist), confirming the semantic validity of our expert modules. Ultimately, NextLocMoE produces distinct and correct next location predictions for the two cases—highlighting its ability to make interpretable and effective forecasts.

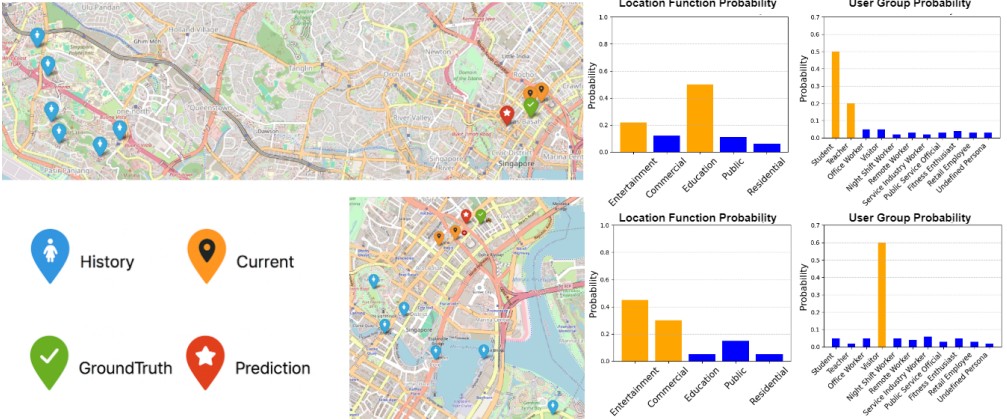

Figure 3: Case study for NextLocMoE.

## 6 CONCLUSION

We propose NextLocMoE, a dual-level Mixture-of-Experts (MoE) enhanced large language model for next location prediction. It incorporates two complementary modules: Location MoE, which captures fine-grained location functional semantics using a fixed top-$k$ expert routing, and Personalized MoE, which models user behavioral patterns diversity via confidence-thresholded dynamic routing. To improve contextual awareness and reliability in expert selection, NextLocMoE introduces a historical-aware router, which explicitly incorporates long-term historical trajectories during expert routing. Empirical results show that NextLocMoE outperforms existing baselines in accuracy, generalization, and inference speed. Case study also shows its interpretability. Nonetheless, Next-LocMoE incurs notable training-time memory costs due to maintaining full FFN sub-networks per user group expert. Future work will explore expert compression techniques, such as weight-splitting from Llama-MoE (Zhu et al., 2024a), to reduce this overhead.

## 7 ETHICS STATEMENT

This work uses three human mobility datasets: Kumamoto, Shanghai and Singapore. All datasets are either publicly released or obtained under formal research agreements that ensure compliance with privacy protection policies, and all datasets are fully anonymized, indexed by non-traceable user IDs. The user group categories used in the Personalized MoE are coarse-grained semantic labels that do not contain any personally identifiable information (e.g., name, age, gender). Our model leverages only these abstract group priors for routing and cannot be used to identify, track, or surveil specific individuals. Therefore, the proposed approach does not pose additional risks of discriminatory profiling or privacy leakage beyond those inherent in anonymized mobility data.

## 8 REPRODUCIBILITY STATEMENT

We have taken several measures to ensure the reproducibility of our results. For code, we provide an anonymized repository containing the full implementation of NextLocMoE, including model architecture, training scripts, and evaluation pipelines. For data, detailed descriptions of the three datasets are included, along with preprocessing procedures and data partition strategies. For hyperparameters, complete hyperparameter settings are listed in Appendix, covering training epochs, learning rates, embedding dimensions, and MoE routing thresholds. Together, these resources allow researchers to fully reproduce our experiments.

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

# A DETAILED RELATED WORK

## A.1 NEXT LOCATION PREDICTION

Next Location prediction aims to forecast the most probable place a user will visit in the near future, based on his/her past mobility trajectory. This task has attracted increasing research interest. Over time, models have evolved significantly to better capture the complex temporal dynamics, spatial semantics, and behavioral diversity inherent in human mobility (Chekol & Fufa, 2022; Rajule et al., 2023; Zhang & Dai, 2018). Broadly, existing methods can be categorized into three major paradigms: RNN-based models that emphasize sequential learning (Sherstinsky, 2020), attention-based models that enhance long-range context integration (Vaswani et al., 2017), and LLM-based models that leverage pretrained knowledge and reasoning capabilities (Achiam et al., 2023; Zhu et al., 2024a). Below, we provide a detailed review of representative methods within each category.

### A.1.1 RNN-BASED NEXT LOCATION

Early approaches to next-location prediction primarily relied on recurrent neural networks, such as GRU (Chung et al., 2014) and LSTM (Graves, 2012), to model sequential dependencies. Deep-Move (Feng et al., 2018) jointly models short-term interests and long-term preferences, capturing user mobility patterns over multiple timescales. SASRM (Zhang et al., 2020) introduces a semantic- and attention-enhanced spatio-temporal recurrent model, which better captures location semantics and contextual dependencies. MCN4Rec (Li et al., 2024b) takes a multi-perspective approach, collaboratively learning from both local and global views to model heterogeneous relationships among users, POIs, temporal factors, and activity types. (Zhang et al., 2022) extend the theoretical foundation of mobility prediction by introducing a new upper bound that incorporates not only sequential patterns but also contextual features such as time and location categories.

In parallel, several models address practical challenges like data privacy and label scarcity. SecureDeepMove (Liu et al., 2024b) integrates secret sharing and secure two-party computation to perform inference without compromising user privacy. SelfMove (Hong et al., 2023a) adopts a self-supervised learning strategy to disentangle time-invariant and time-varying factors, enabling training without labeled next-POI data. (Hasan & Jeong, 2022) design an LSTM-based system that effectively leverages sequential and temporal cues from device-level mobility logs.

Hybrid architectures also emerge. SAB-GNN (Xue et al., 2022) fuses LSTM with a Graph Neural Network to jointly capture spatial dependencies across urban regions and temporal dynamics from user mobility and web search activity. Notably, it incorporates decaying public awareness signals to forecast multiwave patterns in mobility—demonstrating the flexibility of RNN-based frameworks in complex real-world scenarios.

### A.1.2 ATTENTION-BASED NEXT LOCATION PREDICTION

With the rise of the Transformer architecture (Vaswani et al., 2017), attention-based methods have rapidly become the mainstream in next-location prediction due to their superior ability to model long-range dependencies and capture complex spatial-temporal interactions. These models often extend attention mechanisms with auxiliary data, personalized encodings, or graph structures to enhance predictive performance and generalization.

Several works enhance spatial-temporal reasoning via graph-augmented attention. TrajGraph (Zhao et al., 2024) employs a graph Transformer to efficiently encode spatiotemporal context under reduced computational complexity. GETNext (Yang et al., 2022) and SEAGET (Al Hasan & Anwar, 2025) construct trajectory flow graphs to incorporate collaborative mobility signals into attention-based models. AGCL (Rao et al., 2024) introduces a multi-graph learning framework with adaptive POI graphs, spatial-temporal attention, and bias correction. iPCM (Song et al., 2025) combines global trajectory data with personalized user embeddings using a Transformer encoder and probabilistic correction module.

Another line of work explores behavior modeling and user preference learning. MHSA (Hong et al., 2023b) models transition relations among locations using multi-head self-attention. CLLP (Zhou et al., 2024) fuses local and global spatiotemporal contexts to track evolving user interests. CTLE (Lin et al., 2021) maps contextual encodings into a target location embedding, followed

by bidirectional Transformer modeling. MCLP (Sun et al., 2024) leverages topic models to extract latent user preferences and enhances arrival time estimation via attention. FHCRec (Chen et al., 2025) captures both long- and short-term patterns through hierarchical contrastive learning over subsequences. STMGCL (Jia et al., 2023) introduces temporal group contrastive learning within a self-attention encoder to uncover user preference groups.

Auxiliary signals are widely integrated. PRPPA (Liang et al., 2019) combines static user profiles, recent check-in behavior, and temporal point processes into a unified attention framework. San-Move (Wang et al., 2023a) proposes a non-invasive self-attention module that utilizes auxiliary trajectory signals to learn short-term preferences. TCSA-Net (Sun et al., 2022) jointly captures long- and short-term mobility patterns from sparse and irregular trajectories. LoTNext (Xu et al., 2024) addresses the long-tail challenge via graph and loss adjustments that rebalance POI interaction distributions.

Domain-specific and event-aware attention models have also emerged. Physics-ST (Gao et al., 2024) infuses physics priors into human mobility modeling by formulating movement as governed by potential energy dynamics, combined with graph-based attention and temporal correction. (Wang et al., 2023c) incorporates event embeddings to represent both routine behaviors and disruptions. The BERT-based method of (Terashima et al., 2023) repurposes pretrained language encoders for trajectory modeling. (Shukla & Shukla, 2024) uses an encoder–decoder attention structure for coordinate-level prediction.

### A.1.3 LLM-BASED NEXT LOCATION PREDCTION

In recent years, breakthroughs in large language models (LLMs)(Achiam et al., 2023; Liu et al., 2024a; Touvron et al., 2023) have sparked growing interest in their application to next-location prediction. These models offer strong reasoning abilities, contextual understanding, and pre-trained world knowledge that can complement traditional mobility modeling frameworks. Llama-Mob(Tang et al., 2024) and LLMMob (Wang et al., 2023b) incorporate task-specific prompting strategies to adapt LLMs for spatial prediction tasks. Going further, NextLocLLM (Liu et al., 2024c) introduces a dual-role usage of LLMs, functioning as both semantic enhancer and next-location predictor, thereby improving both accuracy and generalization across mobility datasets. AgentMove (Feng et al., 2025) decomposes the next-location prediction task into three specialized components: a spatial-temporal memory module that captures individual behavioral patterns, a world knowledge generator that infers structural and urban influences, and a collective knowledge extractor that models shared mobility patterns across populations. Meanwhile, CausalMob (Yang et al., 2024a) introduces a causality-inspired framework that leverages LLMs to extract latent intention signals tied to external events. It then estimates their causal effects on user mobility while controlling for spatial and temporal confounders—highlighting the potential of LLMs to go beyond pattern recognition and engage in causal reasoning within human mobility modeling.

### A.2 MIXTURE OF EXPERTS

Mixture-of-Experts (MoE) has become a foundational approach for scaling large models while maintaining computational efficiency. Unlike dense models that activate all parameters for every input, MoE architectures route each token or input to a small subset of specialized experts, drastically reducing the per-example computation (Lo et al., 2024). Early works such as GShard (Lepikhin et al., 2020) and Switch Transformer (Fedus et al., 2022) pioneered this direction. GShard introduced a scalable training framework with automatic sharding support, enabling a 600B-parameter Transformer to be trained on 2048 TPUs. Switch Transformer further simplified the routing mechanism by activating only one expert per token, leading to better training stability and communication efficiency, and achieving 7× speedups during pretraining. These foundational designs demonstrate the practicality of scaling models to the trillion-parameter regime without linearly increasing computational cost.

Subsequent works have focused on improving expert specialization, routing flexibility, and deployment efficiency. DeepSeekMoE (Dai et al., 2024) introduces fine-grained expert segmentation and shared experts to encourage non-overlapping expertise and reduce redundancy. PMoE (Jung & Kim, 2024) adopts an asymmetric transformer layout, with shallow layers handling general knowledge

and deep layers using progressively added experts for continual learning, mitigating catastrophic forgetting.

Beyond training from scratch, several methods propose transforming existing dense models into MoE architectures. LLaMA-MoE (Zhu et al., 2024a) partitions feed-forward layers of LLaMA-2 and uses continual pretraining to preserve language capability while introducing sparse expert routing. MoE Jetpack (Zhu et al., 2024b) repurposes dense model checkpoints and introducing a hyperspherical adaptive MoE layer for efficient fine-tuning.

Efficiency during inference and dynamic routing has also been actively explored. (Huang et al., 2024) adjusts the number of active experts per input based on difficulty, dispatching more experts for complex reasoning tasks. (Lu et al., 2024) propose post-training strategies to reduce active parameters per task, improving MoE deployability without retraining. MixLoRA (Shen et al., 2024) adapts MoE to multimodal instruction tuning by constructing instance-specific low-rank LoRA adapters to reduce task interference.

## B    LOCATION FUNCTION NATURAL LANGUAGE DESCRIPTION

We select five location function categories—Entertainment, Commercial, Education, Public Service, and Residential—based on their prevalence and interpretability in urban computing, region representation, POI classification, and trajectory analysis literature. While the granularity and naming may vary across studies, these five categories appear frequently and exhibit strong generalizability. For example, Chen et al. (2023) uses the same five-class scheme as our NextLocMoE for building function classification: residential, commercial, entertainment, public service, and education. Luo et al. (2023) segments the city into residential, commercial, logistics and storage, transportation, green areas and squares (entertainment), and public service. Hong et al. (2023b) categorizes POIs for trajectory prediction as entertainment, residential, schools (education), services and transportation (public service), and shopping (commercial). Ma et al. (2019) clusters areas into entertainment, public service, hotel (residential), education, and food (commercial). (Xiong & Li, 2025) conducts functional clustering of urban spaces, identifying common classes like commercial, tourism (entertainment), residential, public service, and transportation. Table 4 provides natural language descriptions of these predefined semantic categories. Each category reflects a distinct aspect of urban space usage and is used to initialize corresponding experts with LLM-encoded semantic priors.

## C    USER GROUP NATURAL LANGUAGE DESCRIPTION

We define a set of representative user groups based on common mobility behaviors, following the design principle of (JIAWEI et al., 2024), which introduces ten distinct user personas. They argue that while increasing the number of groups improves behavioral diversity, it also compromises efficiency; ten categories strike a balance between representativeness and computational cost. Motivated by this, we adopt the same ten user group categories as expert classes for our Personalized MoE module. Table 5 provides natural language descriptions for the predefined user groups.

## D    PROMPT PREFIX

Fig.4 outlines the specific task and data prompt prefix used in NextLocMoE. The prompt prefix begins by defining the task and providing a detailed description of the dataset structure. Additionally, the Additional Description section emphasizes how to think about this task using the provided data.

## E    DETAILED EXPLANATION OF CROSS-CITY SEMANTIC GENERALIZATION

In this section, we explain the mechanism of NextLocMoE's cross-city semantic generalization. First, NextLocMoE does not treat coordinates themselves as the carriers of urban functional semantics. Normalized coordinates are used purely to provide a unified and comparable spatial scale, enabling the model to process locations from different cities within a consistent spatial range. Semantic meaning is not determined by coordinate values, but by the Location Semantics MoE, whose five function experts correspond to "Commercial", "Residential", "Education", "Entertainment",

Table 4: Location Function Natural Language Description.

| Location Function | Description |
|---|---|
| Entertainment | This category includes scenic spots, sports venues, and recreational facilities, offering activities for leisure, entertainment, and social interactions.Typical examples include amusement parks, cinemas, stadiums, and bars. Users often visit for relaxation, nightlife, sports, and cultural experiences, with peak times in evenings and weekends. |
| Commercial | This category encompasses businesses, financial institutions, automotive services, shopping centers, and dining establishments, supporting daily consumer and professional needs. Typical examples include malls, banks, car dealerships, and restaurants. Users often visit during working hours or weekends for shopping, financial transactions, or dining. |
| Education | This category covers institutions focused on academic, cultural, and scientific learning. Typical examples include schools, universities, libraries, and research centers. Users often visit on weekdays for study, teaching, research, and cultural enrichment. |
| Public Service | This category includes government offices, healthcare facilities, transportation hubs, and other essential public infrastructure. Typical examples include city halls, hospitals, bus stations, and utility centers. Users often visit for administrative tasks, medical needs, commuting, or essential services, with varied peak hours depending on the service type. |
| Residential | This category comprises housing areas, mixed-use developments, and temporary accommodations. Typical examples include apartment complexes, residential neighborhoods, and hotels. Users often visit for long stays, typically peaking in the evenings, weekends, and holidays. |

```
<|start_prompt|>
Task Description:
        Predict the next possible location, in normalized mercator coordinates, of a resident
        based on their historical and current movement trajectory.
Data Description:
        This dataset includes mobility trajectory data of residents.
        Each record consists of historical and current trajectories.
        Historical trajectory contains 40 records, and current trajectory consists of 5 records.
Additional Description:
        Historical trajectory describes travel patterns and frequently visited places,
        while current trajectories reflect user's current location and their short-term travel intentions.
<|end_prompt|>
```

Figure 4: Prompt prefix used in NextLocMoE.

and "Public Service". Each category has its own independent natural-language description, which is encoded using an LLM to initialize the parameters of its corresponding expert. These experts are shared across all cities, ensuring that their semantic directions remain stable and aligned regardless of a city's coordinate system.

Second, NextlocMoE does not infer location semantics directly from coordinates. Instead, semantic assignment is determined by the MoE router, which dynamically selects experts based on the trajectory context. The router takes as input both the initial embedding of the current location and the user's historical behavioral representation, and learns which function experts should be activated for a given mobility pattern. Therefore, even if two locations in different cities share similar normalized coordinates, their routing patterns—and thus location semantics—will differ if their historical trajectory contexts differ. In other words, semantics arise from the combination of location function experts and MoE routing, not from coordinate similarity.

Under this mechanism, cross-city transfer does not rely on aligning coordinate spaces across cities. Instead, it relies on shared and semantically aligned location semantics expert spaces. During training on the source city, the router learns mappings from historical trajectory patterns to function-

Table 5: User Group Natural Language Description.

| User Group | Description |
|---|---|
| Student | This persona represents individuals who typically travel to and from educational institutions at regular times, such as morning arrivals and afternoon departures. Their mobility is highly time-structured and centered around campuses, libraries, and nearby service areas. |
| Teacher | This persona regularly commutes to educational institutions during weekday mornings and returns home in the late afternoon or early evening. Their travel patterns align closely with school schedules, often involving brief visits to nearby commercial or service areas. |
| Office Worker | This persona has a fixed daily commute, traveling to office districts or commercial centers in the morning and returning home in the evening. Their mobility follows a consistent weekday routine with limited variation. |
| Visitor | This persona tends to travel throughout the day with less predictable patterns. They frequently visit tourist attractions, cultural landmarks, dining areas, and shopping districts, especially in central urban zones. |
| Night Shift Worker | This persona often travels outside of standard business hours, especially during late evenings or at night. Common destinations include hospitals, factories, 24-hour service locations, and late-night dining spots. |
| Remote Worker | This persona has non-standard travel patterns. They frequently visit coworking spaces, cafÃ©s, or quiet public environments at various hours of the day, with flexible scheduling that may shift across weekdays. |
| Service Industry Worker | This persona has irregular travel times throughout the day. They frequently move between restaurants, shopping areas, entertainment venues, and other customer-facing POIs, reflecting shift-based work in dynamic urban zones. |
| Public Service Official | This persona often works in rotating shifts, leading to variable travel patterns across different times of the day and night. Common destinations include government offices, transport hubs, hospitals, and administrative centers. |
| Fitness Enthusiast | This persona is active during early mornings, evenings, or weekends. Their mobility revolves around gyms, sports facilities, parks, and wellness-related POIs. Visit durations tend to be regular and intentional. |
| Retail Employee | This persona typically begins travel in the late morning and returns in the evening. Their destination patterns focus on malls, retail stores, and service clusters, reflecting the opening and closing hours of retail operations. |
| Undefined Persona | This persona does not clearly belong to any predefined behavioral category. Their travel patterns may be irregular, spontaneous, or inconsistent across time and location. |

expert combinations. When transferred to a target city, these location semantics experts remain valid and consistent, while the router automatically allocates activation weights based on the given historical trajectories, enabling to produce reasonable semantic interpretations without requiring any labels from the new city.

Finally, our choice of predicting normalized coordinates rather than raw coordinates or location IDs follows naturally from this mechanism. Raw coordinates differ greatly in scale and range across cities, making it difficult for shared experts to learn consistent transformations; location IDs are city-specific and cannot generalize to unseen cities. Normalized coordinates provide a unified geometric scale, allowing the shared experts to apply consistent mappings across cities, while regional semantics are dynamically determined by the MoE router based on mobility behavioral patterns. Therefore, the key to cross-city transfer lies not in coordinate similarity, but in the cross-city consistent semantics encoded by location semantics experts and the router's adaptive expert.

Table 6: Dataset Description.

| Dataset | Num of Records | Time Span (day) | Num of Users | Avg Interval (min) | Num of Locations |
|---------|----------------|-----------------|--------------|--------------------|-----------------|
| Kumamoto | 6696506 | 60 | 17965 | 68.4 | 40000 |
| Shanghai | 1337256 | 8 | 30421 | 94.8 | 10085 |
| Singapore | 2714672 | 31 | 17098 | 61.4 | 4720 |

## F  DATASET DESCRIPTION

We use three real-world mobility datasets to validate the effectiveness of NextLocLoE, and the detailed descriptions of these datasets are as follows:

**Kumamoto** [1] This is an open-source and anonymized dataset of human mobility trajectories from mobile phone location data. The raw dataset was released by Yahoo Japan Corporation and contains four anonymized mobility trajectory sets. Based on spatial distribution and heatmap analysis, these datasets are estimated to correspond to Kobe, Hiroshima, Sapporo, and Kumamoto. We use the Kumamoto dataset as the representative one of the four Yahoo trajectory datasets in our main experiments because it has moderate trajectory density, a clear spatial layout, and well-separated functional zones. The location pings are discretized into 500meters $\times$ 500meters grid cells and the timestamps are rounded up into 30-minute bins.

**Shanghai** [2] This dataset contains mobility records that cover the metropolitan area of Shanghai from April 19 to April 26 in 2016. We selected the core areas of Puxi and the neighborhoods within the Middle Ring Road of Pudong. The location pings are discretized into 200meters $\times$ 200meters grid cells.

**Singapore** This data is collected by one mobile SIM card company in Singapore. It is proprietary and provided under a restricted research agreement with the data owner. We choose the locations in central Singapore. The location pings are discretized into 200meters $\times$ 200meters grid cells. In addition to mobility trajectories, the dataset includes corresponding anonymized demographic attributes (e.g., age, gender, and occupation), which enable us to construct user group labels for validating the interpretability and reliability of our Personalized MoE module.

## G  BASELINE DESCRIPTION

The details of baseline methods are briefly summarized as follows.

- LSTM (Graves, 2012) A type of recurrent neural network capable of learning order dependence in sequence prediction problems.
- GRU (Chung et al., 2014) Similar to LSTMs, GRUs are a streamlined version that use gating mechanisms to control the flow of information and are effective in sequence modeling tasks.
- DeepMove (Feng et al., 2018) This model uses the attention mechanism to combine historical trajectories with current trajectories for prediction.
- SoloPath Anda et al. (2024) It incorporates Time2Vec to capture both periodic and trend-based temporal features and utilizes CatBoost to handle structured, non-sequential trajectory data.
- MHSA (Hong et al., 2023b) An attention-based model that integrates various contextual information from raw location visit sequences.
- CLLP (Zhou et al., 2024) It integrates both local and global spatiotemporal contexts to better capture dynamic user interests.
- GETNext (Yang et al., 2022) It introduces global trajectory flow graphs and graph-enhanced Transformer models.

---

[1]https://zenodo.org/records/13237029
[2]https://github.com/vonfeng/DPLink

- SEAGET (Al Hasan & Anwar, 2025) It uses graph Transformer to leverage collaborative mobility signals to improve predictive performance.
- ROTAN (Feng et al., 2024) It proposes a brand new Time2Rotation technique to capture the temporal information.
- LoTNext (Xu et al., 2024) It addresses data sparsity by enhancing modeling of rare long-tail locations to improve prediction on infrequent places.
- Mobility-LLM (Gong et al., 2024) It explicitly models user visiting intentions and travel preferences, extracting semantic signals from mobility data to boost prediction.
- AgentMove (Feng et al., 2025) A large language model–based agentic framework that leverages reasoning and tool use for zero-shot next location prediction.
- SILO Sun et al. (2025) A semantic integration framework that combines LLMs with multi-source contextual features to strengthen semantic understanding in location prediction.
- LLM4POI (Li et al., 2024a) It effectively uses the abundant contextual information present in LBSN data.
- Llama-Mob (Tang et al., 2024) It instruction tuned Llama for mobility prediction. For alignment, we replace its backbone to Llama3.2-3B, as NextLocMoE uses.
- NextLocLLM (Liu et al., 2024c) It leverages LLM as both a semantic enhancer and a predictor.
- LLmMob (Wang et al., 2023b) It introduces concepts of historical and contextual stays to capture the long-term and short-term dependencies in human mobility.
- ZSNL (Beneduce et al., 2025) It is a purely prompt based model designed for zero-shot next location prediction.

## H  ZERO-SHOT EXPERIMENT SETTING

Here we describe the experiment setting used in our zero-shot experiments. For cross-city models that require training (NextLocMoE, Llama-Mob, and NextLocLLM), we adopt a unified zero-shot setting: the model is trained only on the training set of the source city, and once training is completed, it is evaluated directly on the test set of the target city. No training or validation data from the target city are used, and no fine-tuning or adaptation is performed at any stage. For example, in the Shanghai→Kumamoto setting, all models are trained merely on the Shanghai training set and then evaluated on the Kumamoto test set, ensuring that the comparison reflects strict zero-shot cross-city transferability. For prompt-based methods that do not require training (ZS-NL and LLMMob), we strictly follow their original formulations: since these methods do not involve a training phase, we directly run inference on the target city's test set using their corresponding prompt templates.

## I  FURTHER HYPERPARAMETER SETTINGS

We provide the full hyperparameter list for Kumamoto dataset in Table 7.

## J  ABLATION STUDY

### J.1  ABLATION STUDY ON MOE

To evaluate the contributions of the Location Semantics MoE and Personalized MoE in NextLoc-MoE, we perform ablation studies on Singapore dataset (fully-supervised) and further assess their transferability in Singapore → Kumamoto zero-shot scenario. The results are shown in Table 8. In the fully-supervised setting, removing either module leads to noticeable performance drops. Specifically, discarding the Location Semantics MoE reduces the model's ability to encode multi-functional spatial semantics, while removing the Personalized MoE has an even larger negative effect, confirming the necessity of explicit user persona modeling. In the zero-shot transfer setting, the performance

Table 7: Hyperparameter list for Kumamoto dataset.

| | |
|---|---|
| epoch | 100 |
| beginning learning rate | 0.0001 |
| $L_1$ | 8 |
| $L_2$ | 4 |
| spatial vector dimension | 128 |
| day embedding dimension | 16 |
| hour embedding dimension | 16 |
| duration vector dimension | 16 |
| $M$ | 40 |
| $N$ | 5 |
| $\tau$ | 0.8 |
| $\lambda$ | 300 |

Table 8: Ablation Study on MoE.

| Method | Fully-supervised (Singapore) | | | Zero-shot (Singapore → Kumamoto) | | |
|---|---|---|---|---|---|---|
| | Hit@1 | Hit@5 | Hit@10 | Hit@1 | Hit@5 | Hit@10 |
| NextLocMoE | 9.733% | 34.34% | 40.71% | 15.81% | 34.66% | 47.41% |
| No Location Semantics MoE | 8.827% | 31.54% | 39.27% | 12.03% | 29.47% | 40.09% |
| No Personalized MoE | 8.639% | 30.82% | 38.51% | 11.82% | 18.46% | 38.69% |

Table 9: Ablation Study on History-Aware Router.

| Method | Fully-supervised (Shanghai) | | | Zero-shot (Shanghai → Kumamoto) | | |
|---|---|---|---|---|---|---|
| | Hit@1 | Hit@5 | Hit@10 | Hit@1 | Hit@5 | Hit@10 |
| with $h^{hist}$ | 64.92% | 75.88% | 77.43% | 16.02% | 36.06% | 48.42% |
| without $h^{hist}$ | 60.14% | 70.62% | 72.21% | 13.03% | 27.71% | 41.69% |

gaps widen further. Without the Location Semantics MoE, the model struggles to generalize functional semantics across cities, and without the Personalized MoE, the model becomes highly fragile to unseen user behaviors in new environments. These results validate that both modules are indispensable for achieving robust prediction and effective cross-city generalization.

### J.2 ABLATION ON HISTORY-AWARE ROUTER.

To assess the effectiveness of the history-aware router, we compare model performance with and without incorporating the long-term trajectory representation $h^{hist}$. As shown in Table 9, incorporating the history-aware router consistently improves both fully-supervised and zero-shot performance. With long-term trajectories guiding expert selection, NextLocMoE better captures user preferences and avoids over-reliance on short-term context. This effect is particularly evident in cross-city transfer, where the absence of historical signals leads to unstable routing and weaker generalization.

## K ROBUSTNESS ANALYSIS OF POST-PREDICTION RETRIEVAL

During inference, NextLocMoE employs a KD-Tree as a post-processing retrieval step to map the predicted continuous coordinates to the nearest discrete location IDs within the candidate set of the evaluation city. Since these candidate locations are derived from a regular grid partition, their spatial distribution is uniform and well-structured. As a result, the KD-Tree mapping is deterministic, consistently returning the same nearest location for a given predicted coordinate, which ensures stability and reproducibility. Importantly, the KD-Tree is not involved during training, and thus does not impose any constraint that the next location must be geographically close to the current one. Instead, it simply serves as a practical bridge between continuous outputs and discrete location IDs.

Table 10: Robustness analysis of KD-tree post-processing (in meters)

| Model | Shanghai | Kumamoto | Singapore | Singapore → Kumamoto | Shanghai → Kumamoto |
|---|---|---|---|---|---|
| Llama-Mob | 1146 | 3849 | 2189 | 3446 | 3356 |
| NextLocLLM | 505 | 3070 | 1441 | 2824 | 2791 |
| NextLocMoE | 423 | 2359 | 1021 | 2785 | 2542 |

The robustness of this mapping depends on the distance between the predicted coordinate and the ground-truth location. If the model prediction is close to the ground-truth, the KD-Tree will almost always return the correct discrete ID; if the prediction deviates significantly, the risk of mismatched mapping increases. To quantify this effect, we measure the average Euclidean distance (in meters) between predicted and ground-truth coordinates across multiple datasets and transfer settings. A smaller average distance implies higher robustness, since it reduces the likelihood of mismatched KD-Tree projection. As shown in Table 10, NextLocMoE consistently achieves significantly lower prediction errors compared to Llama-Mob and NextLocLLM across all datasets and transfer settings. This demonstrates that our model outputs are closer to the true positions, thereby reducing the likelihood of mismatched mappings and ultimately enhancing the accuracy, robustness and deployment reliability of our framework.

## L USER GROUP-EXPERT ACTIVATION CONSISTENCY

To evaluate whether the Personalized MoE module can reliably activate experts consistent with true user groups, we measure the user group-expert activation consistency rate on the Singapore dataset. Specifically, for each test user we check whether the ground-truth user group expert is included in the activated expert set. Figure 5 reports the activation consistency rates across the ten predefined user groups. Overall, the module achieves relatively high consistency. Distinctive user groups such as Student and Teacher obtain higher alignment, whereas more heterogeneous groups such as Night Shift Worker and Service Worker are lower. This indicates that the router, guided by both semantic priors and historical trajectory representations, can generally select experts that match user-level behavioral patterns.

Nevertheless, it is important to note that strong user group-expert activation consistency rate does not automatically translate into high next-location prediction accuracy. Alignment ensures that the model activates semantically meaningful experts, but accurate prediction further depends on the quality of trajectory signals. In the Singapore dataset, the long sampling intervals yield sparse trajectories, where large temporal gaps obscure intermediate movements. This makes it uneasy for the model to fully capture short-term transitions, even when the ground-truth persona expert is correctly activated.

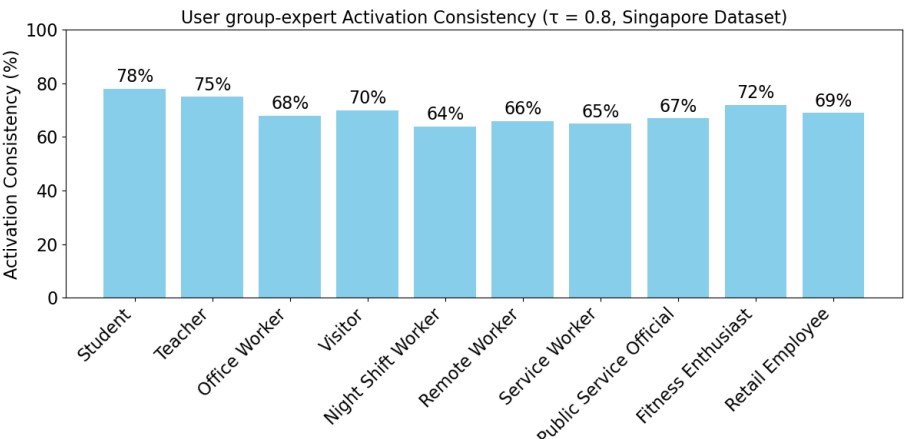

Figure 5: Expert Activation Alignment Rate Across typical user groups

Table 11: Comparison of routing strategies in the Personalized MoE on Singapore dataset.

| Routing Strategy | Hit@1 | Hit@5 | Hit@10 | Inference Time |
|---|---|---|---|---|
| Confidence-threshold | 9.73% | 34.34% | 40.71% | 255s |
| Top-2 | 9.21% | 33.73% | 38.54% | 287s |
| Entropy-based | 9.62% | 34.07% | 40.27% | 256s |

Table 12: Comparison of different history modeling strategies in the router.

| Encoder | Fully-supervised (Kumamoto) | | | | Zero-shot (Singapore → Kumamoto) | | |
|---|---|---|---|---|---|---|---|
| | Hit@1 | Hit@5 | Hit@10 | Inference Time | Hit@1 | Hit@5 | Hit@10 |
| LSTM | 17.01% | 38.45% | 49.58% | 278s (+10s) | 15.02% | 32.06% | 45.14% |
| Attention | 17.57% | 38.91% | 49.82% | 281s (+14s) | 15.67% | 34.23% | 47.05% |
| TCN | 17.77% | 39.19% | 50.28% | 268s | 15.81% | 34.66% | 47.41% |

## M  ROUTING STRATEGY COMPARISON

To further validate the design of the confidence threshold-based expert routing strategy in Personalized MoE, we compare it against two widely-used alternatives: Top-k routing and Entropy-based routing. For a fair comparison, the entropy threshold is tuned so that the average number of activated experts matches that of the confidence-threshold router, while we set $k$ in Top-k to 2. As shown in Table 11, Top-2 routing, which activates two experts for every input, lacks adaptiveness. It introduces irrelevant low-confidence experts for many inputs, injecting noise and incurring unnecessary computation. This leads to performance degradation and increased inference time. Entropy-based routing is adaptive, but its stopping condition is based on entropy rather than confidence. For some inputs with sharp confidence peaks, it may prematurely stop and miss useful experts; for some flatter distributions, it may over-activate noisy experts. This makes it slightly less precise than our confidence-based method.

## N  COMPARISON OF DIFFERENT HISTORY MODELING STRATEGIES

To assess the robustness and suitability of our historical-aware router, we conduct a comparative study of different historical encoding strategies, including LSTM, Self-Attention, and our TCN. As shown in Table 12, TCN achieves higher accuracy and better inference efficiency. We attribute this to the following factors: TCN is well-suited for modeling long-range dependencies in sequential data while avoiding vanishing gradient issues that commonly affect LSTM. While Self-Attention offers high flexibility, its global receptive field may dilute important signals—especially in sparse and noisy mobility sequences. This leads to suboptimal expert routing in practice.

## O  EVALUATION WITH ALTERNATIVE LLM BACKBONES

To further assess the generality and robustness of our framework, we conduct additional experiments with different backbone LLMs. Specifically, we compare Qwen-2.5–3B and LLaMA-3.1–8B against the backbone used in our main experiments (LLaMA-3.2–3B). Table 13 reports results on the Singapore dataset (fully supervised) and the Singapore → Kumamoto zero-shot transfer setting. The results show that performance across backbones is largely comparable, with NextLocMoE maintaining strong effectiveness regardless of the choice of LLM. LLaMA-3.1–8B achieves slightly better accuracy, suggesting that larger backbones can offer marginal accuracy gains. However, these improvements come at the cost of substantially higher inference time, which increases to about 800s when using LLaMA-3.1–8B compared to around 268s with LLaMA-3.2–3B. This efficiency gap highlights the trade-off between accuracy and computational overhead. Considering this balance, we select LLaMA-3.2–3B as the backbone for our main experiments, since it offers the best compromise between predictive accuracy, efficiency, and deployment feasibility. Importantly, the consistent performance across backbones demonstrates that the benefits of NextLocMoE do not depend on a particular LLM, underscoring the robustness and adaptability of our framework.

Table 13: Evaluation with alternative LLM backbones.

| Backbone | Singapore | | | Zero-shot (Singapore → Kumamoto) | | |
|---|---|---|---|---|---|---|
| | Hit@1 | Hit@5 | Hit@10 | Hit@1 | Hit@5 | Hit@10 |
| Qwen-2.5-3B | 9.65% | 34.12% | 38.94% | 15.72% | 34.47% | 46.91% |
| LLaMA-3.1-8B | 9.94% | 35.33% | 42.63% | 15.94% | 35.02% | 48.77% |
| LLaMA-3.2-3B | 9.73% | 34.34% | 40.71% | 15.81% | 34.66% | 47.41% |

## P    HYPERPARAMETER SENSITIVITY

We examine how freezing different numbers of LLM layers affects performance, while keeping the 4 layers integrated with Personalized MoE. As shown in Fig. 6(a), freezing 8 layers yields the best results. Fewer frozen layers lead to poorer generalization, while freezing more than 8 layers degrades performance. This supports prior findings (Skean et al., 2025) that intermediate layers in decoder-only LLMs offer stronger adaptability. Based on this trade-off, we adopt the 8-layer freezing configuration as default.

In addition, we assess NextLocMoE's performance among different history length M. As shown in Fig. 7, increasing M consistently improves performance in both fully-supervised and zero-shot settings, with improvement being steady when M reaches about 40. Following this observation, we set M = 40 as the default, which offers both strong performance and computational efficiency.

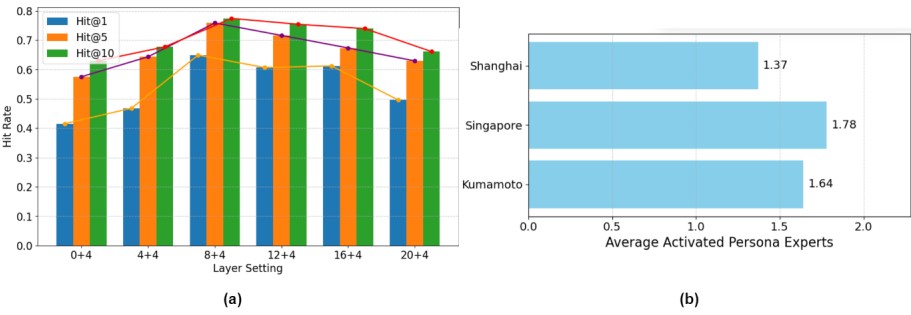

Figure 6: (a) Hyperparameter sensitivity; (b) Personalized expert activation analysis

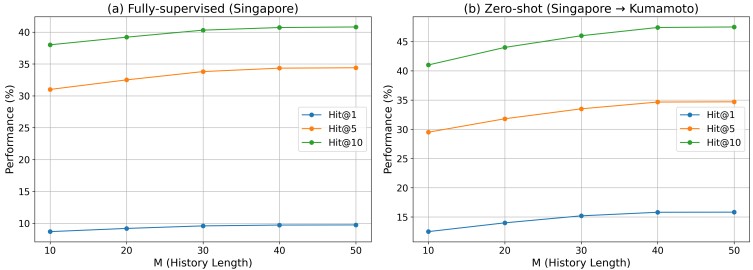

Figure 7: Hyperparameter sensitivity for M

## Q    PERSONALIZED EXPERT ACTIVATION ANALYSIS

We analyze the average number of activated experts in Personalized MoE (Fig. 6(b)). NextLocMoE activates 1.37 experts on Shanghai, 1.78 on Singapore, and 1.64 on Kumamoto—consistently fewer than 2. This shows NextLocMoE's ability to adaptively engage a minimal set of user group experts based on user behavior complexity, ensuring personalized modeling with low inference cost.

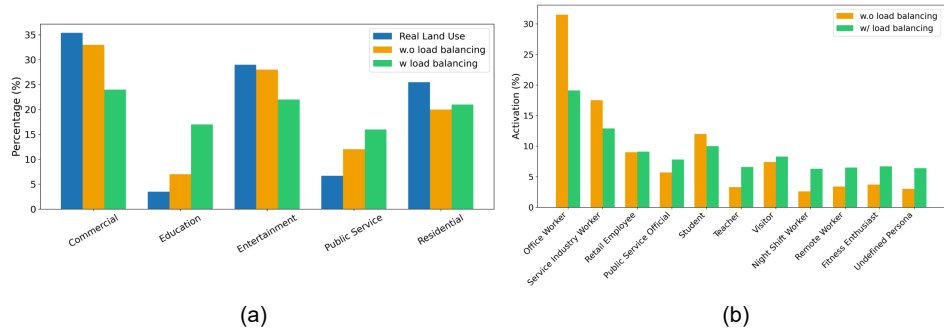

(a)                                      (b)

Figure 8: Influence of load balancing in Sinapore dataset (a) Expert activation distribution of Location Semantics MoE with and without load-balancing loss. (b) Expert activation distribution of Personalized MoE with and without load-balancing loss.

Table 14: Performance with and without load-balancing loss on Location Semantics MoE (Singapore dataset)

|                   | Hit@1   | Hit@5  | Hit@10 |
|-------------------|---------|--------|--------|
| w.o load balancing | 9.733%  | 34.34% | 40.71% |
| w load balancing   | 9.104%  | 32.31% | 38.56% |

## R    INFLUENCE OF LOAD BALANCING IN MoE MODULES

To further examine the impact of load-balancing regularization on the two MoE modules in Next-LocMoE, we conducted experiments comparing model variants with and without the auxiliary load-balancing loss. Following (Fedus et al., 2022; Huang et al., 2024), we adopt the standard auxiliary load-balancing loss that penalizes uneven expert utilization, and we apply it separately to the Location Semantics MoE and the Personalized MoE. Except for the inclusion of load-balancing loss, all other configurations remain identical to ensure fair comparison.

As shown in Table 14 and Table 15, introducing load-balancing loss consistently degrades prediction accuracy on Singapore dataset. Since the only change is the addition of load-balancing loss, this decline indicates that enforcing uniform expert activation disrupts the natural specialization learned by the MoE modules and limits their expressive capacity.

Fig. 8(a) visualizes the expert activation frequencies in the Location Semantics MoE and compares them with the ground-truth land-use distribution in Singapore [3]. Without load-balancing loss, the model naturally learns an imbalanced activation pattern: commercial, entertainment, and residential experts dominate, while education and public service experts are activated less frequently. This pattern aligns with the real-world proportions of the corresponding functional regions, demonstrating that MoE can autonomously capture inherent spatial semantic imbalance. When load-balancing loss is introduced, expert activation becomes significantly more uniform, deviating from the real land-use distribution. This artificial equalization dilutes strong semantic signals associated with high-frequency functional roles and exaggerates low-frequency ones, thereby weakening semantic disambiguation and explaining the observed performance degradation.

We further analyze the activation frequencies of user-behavior experts in the Personalized MoE. NextLocMoE without load-balancing develops a stable and structured pattern of expert utilization: certain experts are activated much more frequently across users, while others remain less active. With load-balancing loss, these activation differences collapse toward uniformity. The behavior-specific expert specialization becomes blurred, reducing the module's ability to disentangle diverse user behavior patterns. This homogenization directly diminishes the representational power of the Personalized MoE and contributes to the decline in predictive accuracy.

---

[3] https://data.gov.sg/datasets?query=land+use&resultId=d_90d86daa5bfaa371668b84fa5f01424f

Table 15: Performance with and without load-balancing loss on Personalized MoE (Singapore dataset)

|  | Hit@1 | Hit@5 | Hit@10 |
|---|---|---|---|
| w.o load balancing | 9.733% | 34.34% | 40.71% |
| w load balancing | 8.807% | 31.57% | 39.04% |

## S  LOCATION SEMANTICS MoE ACTIVATIONS FOR DIFFERENT USER GROUPS

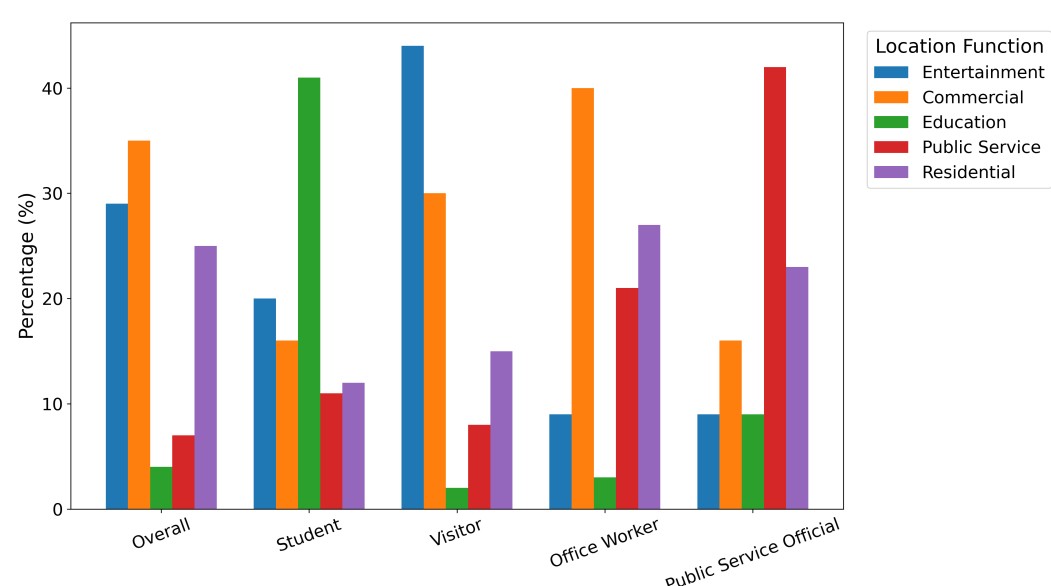

Figure 9: Location Semantics MoE Activations for different user groups

To provide a qualitative understanding of how the Location Semantics MoE specializes across different patterns human mobility, we visualize the activation frequencies of the five location semantic experts for different user groups. We conduct this analysis on the Singapore dataset for four representative user groups: Student, Visitor, Office Worker, and Public Service Official. Fig. 9 reveals clear and interpretable differences across user groups. Students show the highest activation on the Education expert, Visitors predominantly activate the Entertainment expert, Office Workers rely more heavily on the Commercial expert, and Public Service Officials frequently activate the Public Service expert. These patterns are consistent with the expected functional semantics of each group and demonstrate that the experts capture meaningful behavioral regularities. Overall, the observed activation patterns indicate that the Location Semantics MoE does not assign experts arbitrarily. Instead, its routing behavior reflects coherent semantic specialization aligned with real-world mobility patterns, confirming that the module learns an interpretable and functionally grounded decomposition of location semantics.

## T  SENSITIVITY OF NEXTLOCMOE TO THE QUALITY OF LLM SEMANTIC PRIORS

To assess how much the model depends on the quality of the natural-language semantic priors used in the Location Semantics MoE and the Personalized MoE, we conduct several experiments which changes the semantic priors of the two experts on Singapore dataset.

We first replace the full natural-language descriptions with extremely simplified labels ("This location belongs to Entertainment." / "This user group is student."), while keeping all other training settings identical. Under this setting, the performance decreases slightly, but is still effective (as

Table 16: Comparesion between rich labels and simple lables in MoE prior (Singapore dataset)

|  | Hit@1 | Hit@5 | Hit@10 |
|---|---|---|---|
| NextLocMoE | 9.733% | 34.34% | 40.71% |
| Simple label (Location Semantics MoE) | 9.511% | 33.78% | 40.42% |
| Simple label (Personalized MoE) | 9.347% | 33.60% | 40.25% |

Table 17: Ablation study for Location Semantics MoE prior sentences (Singapore dataset)

|  | Hit@1 | Hit@5 | Hit@10 |
|---|---|---|---|
| NextLocMoE | 9.733% | 34.34% | 40.71% |
| w.o S1 | 9.709% | 34.28% | 40.66% |
| w.o S2 | 9.677% | 34.23% | 40.64% |
| w.o S3 | 9.638% | 34.16% | 40.58% |

shown in Table 16). This indicates that richer descriptions provide beneficial semantic structure, yet the overall performance does not rely soly on detailed prompting.

We also conduct a structured ablation of different parts of the textual priors. For Location Semantics MoE, we decompose each description into three components:

- S1, functional definition (e.g., "This category includes scenic spots, sports venues, and recreational facilities...")
- S2, typical examples (e.g., "Typical examples include amusement parks, cinemas, stadiums, and bars.")
- S3, behavior patterns (e.g., "Users often visit for relaxation, nightlife, sports, and cultural experiences...").

For Personalized MoE, we split the persona descriptions into two components:

- S1, identity + primary destination types (e.g., "This persona represents individuals who travel to and from educational institutions at regular times...")
- S2, temporal and behavioral patterns (e.g., "Their mobility is highly time-structured and centered around campuses, libraries, and nearby service areas.")

We remove one component at a time while keeping all other settings unchanged. Across all settings, the removal of any individual semantic component leads to a small decrease in performance. These results indicate that the different parts of the natural-language prior are indeed used by both the routing network and the experts; at the same time, the model remains robust to moderate perturbations of the priors, and does not rely on any single sentence or formulation.

## U    TRAINABLE PARAMETER COUNT

To make the computational overhead of NextLocMoE transparent, we explicitly enumerate and calculate the number of trainable parameters contributed by each component.

### U.1    TRAINABLE PARAMETER COUNT FOR SPATIAL-TEMPORAL EMBEDDING

We first consider the spatial-temporal embeddings. There are four such components:

- A mer2vec embedding matrix of shape $128 \times 2$, contributing 256 parameters.
- A day-of-week embedding of shape $7 \times 16$, contributing 112 parameters.
- An hour-of-day embedding of shape $24 \times 16$, contributing 384 parameters.
- A linear projection for duration with shape $1 \times 16$, contributing 16 parameters.

Table 18: Ablation study for Personalized MoE prior sentences (Singapore dataset)

|  | Hit@1 | Hit@5 | Hit@10 |
|---|---|---|---|
| NextLocMoE | 9.733% | 34.34% | 40.71% |
| w.o S1 | 9.611% | 34.17% | 40.59% |
| w.o S2 | 9.573% | 34.09% | 40.52% |

Summing these terms gives

$$256 + 112 + 384 + 16 = 768$$

trainable parameters for the embedding block.

## U.2 TRAINABLE PARAMETER COUNT FOR TEMPORAL CONVOLUTIONAL NETWORK

The historical encoder is a 5-layer 1D Temporal Convolutional Network. Each TCN layer contains two convolutional blocks (conv1 and conv2) with WeightNorm parameterization and bias.

For each convolution, the weight tensor has shape $176 \times 176 \times 4$, which yields 123,904 weight parameters. In addition, WeightNorm introduces two extra vectors of size 176, and the convolution has a bias term of size 176. Approximating these together as 352 additional parameters, the total parameters per convolutional kernel is 124,256.

Each TCN layer has two such convolutions, so one layer contributes 248,512 parameters. With 5 layers in total, the TCN block contributes

$$5 \times 248,512 = 1,242,560$$

trainable parameters.

## U.3 TRAINABLE PARAMETER COUNT FOR COMPONENTS INSIDE LLAMA-3B

We then account for the trainable parameters inside the Llama-3B backbone. We retain 12 decoder layers, and freeze all attention and FFN weights. Only the LayerNorm parameters and two Personalized MoE layers (inserted among the top four layers in an interleaved manner) remain trainable.

### U.3.1 TRAINABLE PARAMETER COUNT FOR LAYERNORM LAYER

Each decoder layer contains two LayerNorms of size 3072. Thus, each layer contributes 6144 LayerNorm parameters. Across 12 layers, this yields

$$12 \times 6144 = 73,728$$

trainable LayerNorm parameters.

### U.3.2 TRAINABLE PARAMETER COUNT FOR TWO PERSONALIZED MOE LAYERS WITH LORA

Among the 12 decoder layers, two are replaced by Personalized MoE layers. Each MoE layer contains 11 experts, and in each expert we apply LoRA to three linear projections: gate_proj, up_proj, and down_proj. All three projections have input dimension 3072 and output dimension 8192, and we use a LoRA rank $r = 8$.

For one such linear projection with LoRA, the LoRA weights consist of:

- A matrix $A \in R^{3072 \times 8}$, contributing $3072 \times 8 = 24,576$ parameters.
- A matrix $B \in R^{8 \times 8192}$, contributing $8192 \times 8 = 65,536$ parameters.

Thus, each LoRA-augmented linear layer contributes

$$24,576 + 65,536 = 90,112$$

parameters. Since each expert has three such linear layers, gate_proj, up_proj, and down_proj, the total for one expert is

$$3 \times 90,112 = 270,336.$$

There are 11 experts per MoE layer, so all LoRA weights in one MoE layer contribute

$$11 \times 270,336 = 2,973,696$$

parameters.

In addition to the expert-wise LoRA weights, each MoE layer contains a fusion layer and gating components. The fusion layer is a linear projection from 6320 to 3072 dimensions, whose weight matrix has shape $3072 \times 6320$, contributing 19,406,400 parameters, and a bias vector of size 3072, contributing an additional 3072 parameters. Furthermore, a LayerNorm of size 3072 adds weight and bias vectors of size 3072 each, contributing $3072 + 3072 = 6144$ parameters. Finally, the gating head is a linear layer from 3072 to 1, with 3072 weights and 1 bias. Together, these non-expert components in a single MoE layer contribute

$$19,406,400 + 3072 + 6144 + 3073 = 19,418,689$$

parameters. Therefore, the total number of trainable parameters in one Personalized MoE layer is

$$2,973,696 + 19,418,689 = 22,392,385.$$

Since there are two such MoE layers in the model, the overall contribution from MoE + LoRA components is

$$2 \times 22,392,385 = 44,784,770.$$

### U.4    TRAINABLE PARAMETER COUNT FOR OUTPUT AND PROJECTION LAYERS

We also train several lightweight MLP layers used for mapping location functions, user group features and trajectory embedding into the LLM representation space and for producing prediction heads. These output-side components gives 1,529,733 trainable parameters.

### U.5    TOTAL TRAINABLE PARAMETERS

The overall number of trainable parameters in NextLocMoE is therefore

$$768 + 1,242,560 + 44,858,498 + 1,529,733 = 47,631,559,$$

which we round to 47.6M trainable parameters. This corresponds to roughly 1.5% of the 3B parameters of the frozen Llama backbone, confirming that NextLocMoE is a lightweight adaptation rather than a full-scale fine-tuning of the underlying LLM.

## V    LLM USAGE STATEMENT

Apart from the fact that our proposed NextLocMoE itself leverages large language models (LLMs) as its core framework, we merely used LLMs in a limited way to polish the writing style and improve the clarity of exposition. No new research content, results, or scientific insights were generated by LLMs; all conceptual contributions, experimental designs, and analyses are solely attributable to the authors.

