# OpenReview forum: "NextLocMoE: Enhancing Next Location Prediction via Location-Semantics Mixture-of-Experts and Personalized Mixture-of-Experts"
_ICLR.cc/2026/Conference — Submitted to ICLR 2026_

### Official Review · Reviewer_WfzH · 2025-10-23

**Soundness:** 4
**Presentation:** 4
**Contribution:** 3
**Rating:** 8
**Confidence:** 3

**Summary:**

NextLocMoE is a next-generation location prediction framework based on large language models (LLMs). Its core innovation lies in integrating a dual-level Mixture-of-Experts (MoE) architecture, which aims to address the shortcomings of existing location prediction methods in capturing multi-dimensional semantics of locations and diverse behavioral patterns of user groups. Meanwhile, it enhances the routing stability, reliability, and generalization ability of the model.

**Strengths:**

1.The paper applies the MoE architecture to location semantics and user behavior in a hierarchical manner. It not only solves the problem of capturing multi-functional semantics of locations but also realizes the refined distinction of user group behaviors. This dual-level MoE collaborative modeling approach is innovative.

2.The paper initializes expert parameters through natural language descriptions encoded by LLMs, which makes the semantic positioning of expert modules clearer. The expert activation of the Personalized MoE is highly aligned with the real user groups, thus achieving a certain degree of interpretability.

3.The experiments in the paper are sufficient and comprehensive, and the provided code seems correct, which facilitates subsequent research work.

**Weaknesses:**

1.Both the 10 user groups in the Personalized MoE and the 5 function categories in the Location Semantics MoE are manually predefined, lacking the ability of adaptive adjustment in terms of division granularity and category selection. If there are uncovered user groups or location functions in practical scenarios, it may lead to expert activation bias and affect the prediction performance.

2.Although the Personalized MoE does not rely on user IDs, it still needs to guide expert selection through historical trajectory encoding. For new users with no historical trajectories at all (cold-start scenario), the historical-aware router cannot provide effective context.

**Questions:**

1.We note that although NextLocMoE achieves excellent performance, there is still significant room for performance improvement. Is this related to the "Undefined Persona"?

2.Can the length of users' historical trajectories be counted in the provided datasets? And do all users have historical trajectories? How should the cold-start problem be considered?

---

> ### Author Response · Authors · 2025-11-20
> **Response to Reviewer WfzH (1/3)**
>
> >W1: Both the 10 user groups in the Personalized MoE and the 5 function categories in the Location Semantics MoE are manually predefined, lacking the ability of adaptive adjustment in terms of division granularity and category selection. If there are uncovered user groups or location functions in practical scenarios, it may lead to expert activation bias and affect the prediction performance.
>
>
> Thank you for the thoughtful comments. We provide the following clarification regarding the use of predefined categories.
>
> First, the location-function categories and user groups used in our framework are **not arbitrary choices**. They are based on **well-established and widely adopted taxonomies** in urban computing, mobility modeling, and urban functional zone analysis. The functional categories align with mainstream classifications in POI function analysis, functional-zone detection, and transportation behavior research [1–5]. The user groups strictly follow the design of [6], where the authors propose the same ten representative and high-coverage user groups to describe human mobility. Therefore, these categories represent **validated semantic abstractions rather than task-specific heuristics**, and possess **strong generality across cities**.
>
> Second, it is important to emphasize that these predefined semantics **do not act as rigid, hard-coded structures** in our model. Instead, they participate in MoE routing in a **compositional and flexible manner**. A user’s behavioral pattern may activate multiple user group experts simultaneously, and a location’s functional semantics may be expressed through a weighted combination of several location-function experts. Such compositionality allows the model to represent **finer-grained semantics** beyond the original categories. Our predefined categories primarily provide **interpretable semantic anchors** for experts, without restricting the model's expressive capacity.
>
> Third, regarding the reviewer’s concern about **uncovered user groups**, Personalized MoE includes an explicit **Undefined Persona expert**. This expert is designed to handle the trajectories whose behavioral patterns truly cannot be explained by any expert combination. Empirically, its activation rate remains consistently low in our experiments, indicating that the model can generally represent mobility behaviors through combinations of the ten user group experts, while the Undefined expert acts only as a ‘fallback’.
>
> [1] Chen J, et al. Exploring Urban Functional Zones Based on Multi-source Semantic Knowledge and Cross-modal Network. The International Archives of the Photogrammetry, Remote Sensing and Spatial Information Sciences 2023.
>
> [2] Luo G,et al. Urban Functional Zone Classification Based on POI Data and Machine Learning. Sustainability 2023.
>
> [3] Hong Y, et al. Context-aware Multi-head Self-attentional Neural Network Model for Next Location Prediction. Transportation Research Part C: Emerging Technologies 2023.
>
> [4] Ma H, et al. Investigating Road-Constrained Spatial Distributions and Semantic Attractiveness for Area of Interest. Sustainability 2019.
>
> [5] Xiong Y, Li G. Correlation Characteristics Between Urban Fires and Urban Functional Spaces: A Study Based on POI Data and Ripley’s K-Function. ISPRS International Journal of Geo-Information 2025.
>
> [6] Wang J, Jiang R, Yang C, et al. Large language models as urban residents: An llm agent framework for personal mobility generation[J]. Advances in Neural Information Processing Systems, 2024, 37: 124547-124574.

---

> > ### Author Response · Authors · 2025-11-20
> > **Response to Reviewer WfzH (2/3)**
> >
> > >W2: Although the Personalized MoE does not rely on user IDs, it still needs to guide expert selection through historical trajectory encoding. For new users with no historical trajectories at all (cold-start scenario), the historical-aware router cannot provide effective context.
> >
> > Thank you for the insightful comment. We would like to clarify the task definition and the meaning of cold start in our work.
> >
> > First, the task studied in this paper is the **standard next-location prediction** problem. By definition, this task assumes that model receives the user’s historical mobility trajectory and aims to predict the most probable next location. This assumption is consistent across prior work: all existing methods require that the user has **observed mobility records**, so that the model can infer short-term intent and long-term mobility preferences from behavioral history [1–4]. Therefore, the scenario described by the reviewer—a new user with absolutely no historical trajectory—**does not fall within** the scope of next-location prediction and is not the setting targeted in this paper.
> >
> > Second, the “**cold-start scenarios with unseen users**” discussed in our paper refer specifically to splitting the dataset by users, where **test users do not appear in the training set**. In such settings, methods that rely on user static profiles or user embeddings struggle because their learned representations cannot transfer to unseen users. In contrast, Personalized MoE does not use any user information. Its expert routing is driven by user’s historical mobility. Handling users with absolutely no historical records corresponds to a different scenario that **does not fall within** the cold-start setting considered in this paper.
> >
> > [1] Hong Y, Zhang Y, Schindler K, et al. Context-aware multi-head self-attentional neural network model for next location prediction[J]. Transportation Research Part C: Emerging Technologies, 2023, 156: 104315.
> >
> > [2] Feng J, Li Y, Zhang C, et al. Deepmove: Predicting human mobility with attentional recurrent networks[C]//Proceedings of the 2018 world wide web conference. 2018: 1459-1468.
> >
> > [3] Yang S, Liu J, Zhao K. GETNext: Trajectory flow map enhanced transformer for next POI recommendation[C]//Proceedings of the 45th International ACM SIGIR Conference on research and development in information retrieval. 2022: 1144-1153.
> >
> > [4] Wang X, Fang M, Zeng Z, et al. Where would i go next? large language models as human mobility predictors[J]. arXiv preprint arXiv:2308.15197, 2023.
> >
> > > Q1: We note that although NextLocMoE achieves excellent performance, there is still significant room for performance improvement. Is this related to the "Undefined Persona"?
> >
> > Thank you for the interesting question and for acknowledging the overall performance of NextLocMoE. We provide the following clarification on whether the “Undefined Persona” might limit the model’s upper-bound performance.
> >
> > First, from the perspective of model behavior, the **Undefined Persona expert is activated at the lowest rate among all experts**, as shown in our supplementary analysis in Appendix R. The Undefined Persona serves merely as a ‘fallback’ rather than a major contributor to prediction. Consequently, it is **unlikely** to be the factor limiting the overall performance.
> >
> > Second, we think that the performance gap is attributable to **limitations of the task and the data**, rather than to any constraint imposed by the Undefined Persona:
> >
> > (1) The datasets used in this study exhibit long sampling intervals, where multiple unobserved movements might occur between consecutive records. This makes some short-term behavioral intent inherently difficult to infer and lowers the predictability.
> >
> > (2) NextLocMoE currently relies solely on mobility trajectories and does not incorporate external contextual signals such as weather, holidays, traffic conditions, or large-scale events, all of which might have impact on human movement.
> >
> > (3) Prior work shows that human mobility itself has intrinsic limits of predictability [1,2]; even with perfect models and complete data, a fraction of movements remains random or unpredictable.
> >
> > For these reasons, the performance ceiling of NextLocMoE is shaped **far more** by the nature of the task and the data than by the Undefined Persona expert. We believe that incorporating richer external contextual signals and modeling behavioral structures at a finer granularity will provide additional opportunities to further improve the model’s overall performance in future work.
> >
> > [1] Song C, Qu Z, Blumm N, et al. Limits of predictability in human mobility[J]. Science, 2010, 327(5968): 1018-1021.
> >
> > [2] Zhang C, Zhao K, Chen M. Beyond the limits of predictability in human mobility prediction: Context-transition predictability[J]. IEEE Transactions on Knowledge and Data Engineering, 2022, 35(5): 4514-4526.

---

> ### Author Response · Authors · 2025-11-20
> **Response to Reviewer WfzH (3/3)**
>
> >Q2: Can the length of users' historical trajectories be counted in the provided datasets? And do all users have historical trajectories? How should the cold-start problem be considered?
>
> Thank you for the reviewer’s suggestion. We provide the statistics of the raw trajectory lengths in our experiments.
>
> |          | Min  | Mean  | Max   |
> | -------- | ---- | ----- | ----- |
> | Kumamoto | 111  | 1027  | 2297  |
> | Shanghai | 26   | 92    | 190   |
> | Singapore| 74   | 158   | 322   |
>
> These statistics describe the density of the original data, but they do not directly correspond to the model input. Following prior work [1,2], we adopt a unified sliding-window structure during training and evaluation: **each** sample consists of 40 historical records and 5 current records, ensuring consistent input format and providing stable context for the router.
>
> Regarding the cold-start issue, we clarify that the “cold-start scenarios with unseen users” in our paper refer to **user-level splits**, where **users in the test set do not appear in the training set**. Methods relying on user profiles or user embeddings usually fail in this setting because their representations cannot transfer. In contrast, NextLocMoE does not rely on user IDs or any static user attributes. Its expert routing is entirely driven by the user’s historical trajectories. Therefore, under our task definition, NextLocMoE does not suffer from additional user-level cold-start difficulties.
>
> We also include sensitivity experiments on the historical window length M in appendix P. The results show that using a short history (e.g., M=10 or 20) leads to a little bit suboptimal performance, indicating that the router requires richer historical context. As M increases, performance stabilizes around 40, and the improvement from M=40 to M=50 becomes negligible. This suggests that M=40 provides a good balance between capturing sufficient spatiotemporal structure and avoiding the noise introduced by overly long sequences.
>
> [1] Wang X, Fang M, Zeng Z, et al. Where would i go next? large language models as human mobility predictors[J]. arXiv preprint arXiv:2308.15197, 2023.
>
> [2] Luo Y, Liu Q, Liu Z. Stan: Spatio-temporal attention network for next location recommendation[C]//Proceedings of the web conference 2021. 2021: 2177-2185.

---

> ### Author Response · Authors · 2025-11-27
> **Gentle Reminder**
>
> Dear Reviewer WfzH,
>
> We sincerely thank you again for your valuable comments and constructive suggestions. As discussion stage is approaching its end, we would appreciate it if you could please let us know whether our responses and the revised manuscript have addressed your concerns, and let us know if any issues remain.
>
> Best regards,
>
> Author of Submission 86

---

### Official Review · Reviewer_hP3j · 2025-10-25

**Soundness:** 2
**Presentation:** 3
**Contribution:** 2
**Rating:** 2
**Confidence:** 5

**Summary:**

This paper proposes NextLocMoE, a dual-level Mixture-of-Experts (MoE) enhanced LLM framework for next location prediction. The framework integrates two complementary MoE modules:
(1) Location Semantics MoE: Models the multi-functional semantics of locations (e.g., commercial, educational, entertainment) using a fixed top-k expert routing strategy.

(2) Personalized MoE: Captures diverse user behavioral patterns via a confidence threshold-based dynamic routing strategy, enabling group-level personalization without explicit user IDs.

Additionally, the authors introduce a historical-aware router that incorporates long-term historical trajectories into expert selection, enhancing contextual stability and reliability. Extensive experiments on three real-world mobility datasets demonstrate that NextLocMoE consistently outperforms state-of-the-art baselines under both fully-supervised and zero-shot cross-city settings, while achieving significant inference speedups (e.g., 600× faster than Llama-Mob).

**Strengths:**

1. Extensive experiments and detailed analyses (e.g., routing strategies, historical encoding, alternative backbones) demonstrate the robustness and generality of the approach.

2. The writing is clear, and the methodology is described in sufficient detail.

3. The model not only improves accuracy but also drastically reduces inference time, making it suitable for large-scale applications.

**Weaknesses:**

1. The research problem is too well-studied.

2. The Personalized MoE relies on predefined user group descriptions. While LLM-encoded priors help, this may not cover all behavioral patterns in diverse populations.

3. Although the model outperforms baselines in zero-shot transfer, absolute performance (e.g., Hit@1 ~16% on Kumamoto) remains low, indicating room for improvement in cross-city generalization.

**Questions:**

The Personalized MoE uses 10 predefined user groups. Have the authors considered using unsupervised clustering or dynamic group discovery to reduce reliance on predefined categories?

How does the Personalized MoE handle users with ambiguous or mixed behaviors (e.g., the "Undefined Persona" category)? Is there a risk of over- or under-activating experts for such users?

---

> ### Author Response · Authors · 2025-11-20
> **Response to Reviewer hP3j (1/3)**
>
> >W1 The research problem is too well-studied.
>
> While we acknowledge that next-location prediction has been studied extensively, we believe the field is still **far from** being “solved”. Several fundamental challenges remain without mature solutions, and these challenges directly motivate our work.
>
> First, real-world urban locations often carry multiple functional roles simultaneously. Effectively modeling such **multi-semantic locations** remains difficult. Most existing methods rely on a single embedding, which easily leads to semantic compression in mixed-function areas and limits both representation capacity and interpretability.
>
> Second, human mobility patterns exhibit **strong behavioral heterogeneity** across user groups. However, mainstream approaches typically fit all users using a single shared set of parameters, making it difficult to preserve clear behavioral distinctions. As a result, mobility patterns from different user groups are blended together in the learned representations, reducing the model’s ability to specialize or interpret group-specific mobility behaviors.
>
> Third, in the more challenging **cross-city** prediction scenario, existing models commonly suffer substantial performance degradation, and user-level cold start makes approaches relying on user embeddings or static profiles suboptimal. **As the reviewer noted**, although our method outperforms all baselines on Kumamoto zero-shot task, the absolute Hit@1 still has room for improvement (the comment in W3), highlighting that the task **remains intrinsically challenging** rather than mature.
>
> In summary, despite extensive prior work, the core difficulties of next-location prediction, namely multi-semantic locations, mobility behavior heterogeneity, and cross-city generalization, **remain challenging and deserve further research**. We therefore believe that this research direction is **far from** over-studied or matured; instead, it is at a stage where new methodological advances are both necessary and impactful.
>
> > W2: The Personalized MoE relies on predefined user group descriptions. While LLM-encoded priors help, this may not cover all behavioral patterns in diverse populations.
>
> We would like to clarify the design intention and actual behavior of Personalized MoE.
>
> First, the user groups used in our framework are **not** arbitrary categories. They strictly follow the design principles of [1], where these groups serve as **representative behavioral prototypes** capturing the most common and representative mobility patterns observed in real cities.
>
> Second, the routing mechanism of Personalized MoE enables a **compositional representation** of user behavior. The model does not force a user trajectory into any single user group; instead, it dynamically activates multiple experts based on the user’s historical behavior and constructs the final representation through a weighted combination. In other words, a user’s mobility pattern can be expressed as a collaboration of several user-group experts rather than being determined by one fixed category. This enables the model to naturally accommodate typical and mixed mobility patterns.
>
> Furthermore, to address the reviewer’s concern regarding behavioral patterns that do not match any existing user groups, we include an **Undefined Persona expert** in Personalized MoE. As shown in our supplementary experiments (Appendx R), the activation rate of this expert remains consistently low, indicating that the model can generally represent mobility behaviors through combinations of the ten user group experts. The Undefined expert only serves as a ‘fallback‘ when a pattern truly cannot be explained by any expert combination. This design ensures robustness when encountering unfamiliar behaviors while avoiding distorted activation patterns or unintended expert overuse.
>
> [1] Wang J, Jiang R, Yang C, et al. Large language models as urban residents: An llm agent framework for personal mobility generation[J]. Advances in Neural Information Processing Systems, 2024, 37: 124547-124574.

---

> > ### Author Response · Authors · 2025-11-20
> > **Response to Reviewer hP3j (2/3)**
> >
> > > W3: Although the model outperforms baselines in zero-shot transfer, absolute performance (e.g., Hit@1 ~16% on Kumamoto) remains low, indicating room for improvement in cross-city generalization.
> >
> > We appreciate reviewer’s attention to the zero-shot transfer results and *their recognition of our NextLocMoE’s performance advantages*. It is true that, even though NextLocMoE consistently **surpasses** all strong baselines across all metrics, the Hit@1 in the zero-shot setting remains relatively low. We emphasize that this reflects the **inherent difficulty** of the task rather than a limitation of our model, and further highlights the substantial **room for future progress**.
> >
> > First, the real-world mobility datasets used in this study exhibit **long sampling intervals**. Unobserved movements might occur between two consecutive records, making short-term behavioral intentions difficult to infer from visible data. This sparsity fundamentally lowers the predictability, and its impact becomes even more pronounced in zero-shot scenarios.
> >
> > Second, under the strict zero-shot setting, the model has no access to the trajectory records, spatial layouts, or mobility patterns from target cities, and cannot perform any form of domain adaptation. Because cities differ substantially in functional structures and behavioral distributions, such cross-city transfer inevitably suffers from strong **distribution shift**, making it unrealistic for absolute performance to approach that of supervised single-city prediction.
> >
> > Moreover, prior studies have shown that human mobility exhibits **intrinsic limits of predictability** [1,2]. Even within a single city, a portion of movement is inherently random and cannot be inferred from historical trajectories. In cross-city transfer without target-city data, this unpredictable component becomes further amplified, increasing the difficulty of the task.
> >
> > Given these constraints, the relatively low absolute performance in zero-shot settings is **expected**. Nevertheless, NextLocMoE **still consistently outperforms** all strong baselines under identical conditions, indicating its substantial advantage in capturing cross-city consistent location semantics and behavioral patterns. We agree with the reviewer that there remains significant room for improvement. **Most importantly**, this precisely demonstrates that next location prediction is **far from a solved problem (the comment in W1)**, and continues to be an important and challenging research direction.
> >
> > [1] Song C, Qu Z, Blumm N, et al. Limits of predictability in human mobility[J]. Science, 2010, 327(5968): 1018-1021.
> >
> > [2] Zhang C, Zhao K, Chen M. Beyond the limits of predictability in human mobility prediction: Context-transition predictability[J]. IEEE Transactions on Knowledge and Data Engineering, 2022, 35(5): 4514-4526.

---

> > > ### Author Response · Authors · 2025-11-20
> > > **Response to Reviewer hP3j (3/3)**
> > >
> > > > Q1: The Personalized MoE uses 10 predefined user groups. Have the authors considered using unsupervised clustering or dynamic group discovery to reduce reliance on predefined categories?
> > >
> > > Here we clarify the design choice of using predefined user groups in Personalized MoE.
> > >
> > > First, the user groups used in our framework are **not arbitrarily choices**. They are adopted directly from prior work in urban computing [1], where these user groups represent the most common and representative mobility patterns in real cities. These groups serve as **semantically meaningful and cross-city consistent behavioral prototypes** for Personalized MoE.
> > >
> > > Second, the routing mechanism of Personalized MoE enables a **compositional representation** of user behavior. The model does not force a user trajectory into any single user group; instead, it dynamically activates multiple experts based on the user’s historical behavior and constructs the final representation through a weighted combination. This allows mobility patterns to be represented through flexible combinations of multiple behavioral prototypes. Such compositional representations provide **high expressiveness while maintaining interpretability**, without restricting the model to any single predefined user group.
> > >
> > > In contrast, fully unsupervised clustering or dynamic group discovery can indeed uncover latent structures, but the resulting clusters typically **lack clear semantic meaning and are highly sensitive to the data distribution of each city**. Consequently, such clusters often fail to maintain consistent semantics across different cities, making them difficult to transfer to new urban environments and unsuitable for providing **stable behavioral semantics** in cross-city prediction.
> > >
> > > Therefore, predefined user groups offer **consistent behavioral semantics across cities**, while the MoE’s activation mechanism provides sufficient modeling flexibility. Together, they form a desirable balance between **interpretability, robustness, and cross-city generalization**. However, we agree that unsupervised group discovery is a promising direction; future work may explore integrating data-driven structures with semantic user groups to further enhance cross-city semantic alignment.
> > >
> > > [1] Wang J, Jiang R, Yang C, et al. Large language models as urban residents: An llm agent framework for personal mobility generation[J]. Advances in Neural Information Processing Systems, 2024, 37: 124547-124574.
> > >
> > >
> > > > Q2: How does the Personalized MoE handle users with ambiguous or mixed behaviors (e.g., the "Undefined Persona" category)? Is there a risk of over- or under-activating experts for such users?
> > >
> > > First, Personalized MoE handles mixed behaviors through its routing mechanism. Instead of assigning a user to one persona, the router computes a weighted combination of multiple experts based on the user’s historical trajectory. This allows mixed or ambiguous mobility patterns to be represented compositionally. As shown in Fig. 6(b), the average number of activated experts across the three datasets is 1.37 / 1.64 / 1.78, indicating that mixed behaviors are naturally captured through **compositional expert activation**, and thus there is **no risk** of under-activation for such users.
> > >
> > > Second, the Undefined Persona expert is **not** intended to handle all mixed-behavior users. Instead, it serves as a ‘fallback’ only for the very small subset of trajectories truly not explainable by any expert combination. Empirically, its activation rate is only about 3% on Singapore dataset, lower than other experts (Appendix R). This demonstrates that the router generally maps user behaviors to clear semantic combinations and does **not** rely heavily on Undefined Persona expert—thus **avoiding** the risk of over-activation.

---

> ### Author Response · Authors · 2025-11-27
> **Gentle Reminder**
>
> Dear Reviewer hP3j,
>
> We sincerely thank you again for your valuable comments and constructive suggestions. As discussion stage is approaching its end, we would appreciate it if you could please let us know whether our responses and the revised manuscript have addressed your concerns, and let us know if any issues remain.
>
> Best regards,
>
> Author of Submission 86

---

> > ### Comment · Reviewer_hP3j · 2025-11-28
> >
> > Thanks for the response from the authors. After reading the rebuttal and the reviews from the other reviewers. I've raised my rating to 4.

---

### Official Review · Reviewer_Eynx · 2025-10-29

**Soundness:** 3
**Presentation:** 3
**Contribution:** 3
**Rating:** 4
**Confidence:** 2

**Summary:**

This paper proposes NextLocMoE, an LLM-based next-location prediction framework that novelly introduces two complementary MoE modules: a Location Semantics MoE to model multi-functional semantics of locations, and a Personalized MoE to capture user behavioral heterogeneity. The authors also design a history-aware router that uses a TCN of long-term history to stabilize expert selection. The model outputs continuous coordinates and converts coordinates to discrete IDs at inference via KD-Tree retrieval. Experiments on three city datasets show improved Hit@k and much faster inference than LLM-based baselines; extensive appendices provide ablations and robustness checks.

**Strengths:**

1. Clearly articulate two practical failure modes of existing systems (single semantic embedding for locations; single shared model for all users)

2. Innovatively integrating MoE for prediction. Location Semantics MoE gives multiple function-aware embeddings per place; Personalized MoE provides group-level personalization without user IDs — both are interpretable and demonstrated via case studies.

3. Many ablations show the individual contributions of the modules.

**Weaknesses:**

1. The authors claim an interesting argument that omitting the load-balancing loss allows the model to better capture the natural, imbalanced distribution of urban activities.  However, the paper lacks the qualitative experiments needed to validate it.

2. The framework has a high dependence on a fixed set of five location functions and ten user groups. This raises concerns about its generalizability and scalability. It is unclear how well these pre-defined categories would transfer to new cities with different functional layouts (e.g., an industrial city vs. a tourist city).

3. Complex model. The system integrates multiple components (LLM, TCN, two MoEs, LoRA fine-tuning, and KD-Tree). It brings an extremely high cost, which may lead to problems such as difficulties in adjusting hyperparameters and reduced reproducibility.

**Questions:**

1. Could the authors provide a qualitative analysis of the expert activation frequencies? For example, showing the activation rates of the five location experts for a specific type of user trajectory.

2. Are these fixed categories still applicable when the model is applied to a new environment with a different urban layout (e.g., an industrial vs. a tourist city) or different cultural mobility patterns?

3. How much does the quality of LLM semantic priors affect performance? Have you experimented with comparing the results (Detailed, rich descriptions vs. using simple labels (e.g., just "Commercial" or "Student")) for semantic priors? Or the influence of a slight perturbation of semantic priors on the results?

4. Considering the overall complexity of the model, have you ever explored simplifying the model? For instance, if only one of the MoE modules is used, or if a simpler historical encoder (rather than a TCN) is used to boot the router, can a competitive balance be achieved between performance and complexity?

---

> ### Author Response · Authors · 2025-11-20
> **Response to Reviewer Eynx (1/4)**
>
> > W1: The authors claim an interesting argument that omitting the load-balancing loss allows the model to better capture the natural, imbalanced distribution of urban activities. However, the paper lacks the qualitative experiments needed to validate it.
>
> We thank the reviewer for the valuable comments. Following the suggestion, we added a comprehensive comparison between models **with and without load-balancing**, including analyses of **predictive performance** and **expert activations**. All experiments adopt the widely used auxiliary load-balancing loss [1][2][3]. Complete results are presented in Appendix R.
>
> Regarding **predictive performance**, we introduce load-balancing into either Location Semantics MoE or Personalized MoE for controlled comparison. On Singapore dataset, **adding load-balancing consistently leads to performance degradation**. This indicates that load-balancing interferes with the model’s natural expert specialization, making it difficult for MoE to capture inherently imbalanced distributions of location functions and user behavior that exist in real urban environments, ultimately harming predictive accuracy.
>
> To better understand this phenomenon, we further examine the **expert activation distribution in Location Semantics MoE** and compare it with the true land-use proportions in Singapore (Figure 8 (a)). Without load-balancing, experts corresponding to Commercial, Entertainment, and Residential are activated more frequently, whereas Education and Public Service experts are activated less often. Such pattern closely aligned with the real-world land-use. This demonstrates that NextLocMoE is able to capture the intrinsic functional imbalance of urban locations. In contrast, when load-balancing is introduced, the activation frequencies of all experts become forcibly flattened, diverging substantially from the true land-use distribution. As a result, contributions of high-prevalence functions are suppressed, low-prevalence functions are amplified, and the original semantic specialization collapses, ultimately leading to degraded performance.
>
> We observe a **similar pattern in Personalized MoE** (Figure 8(b)). Without load-balancing, user-group experts show clear activation imbalance. After introducing load-balancing, these differences are significantly compressed and the experts are activated at nearly uniform frequencies. This forced equalization removes the data-driven differences in expert usage, making expert roles **less distinct** and limiting Personalized MoE’s ability to use expert activation patterns to differentiate user behaviors. Consequently, both the model’s predictive performance and its interpretability are negatively affected.
>
> Overall, the newly added experiments reinforce our core conclusion: **urban location semantics and user mobility behaviors are inherently imbalanced, and this imbalance itself carries essential information**. Without load-balancing, MoE can naturally exploit these real-world imbalances to effectively capture location semantic and user behavioral. In contrast, load-balancing introduces an inductive bias that is misaligned with the task, forcibly equalizing expert activation, weakening expert specialization, and ultimately reducing predictive accuracy. We have included the corresponding figures and analyses in Appendix R. We sincerely thank the reviewer for the constructive suggestion, which allowed us to present this key design in a more thorough and clearer manner.
>
> | Location Semantics MoE | Hit@1   | Hit@5   | Hit@10  |
> |----------------------|---------|---------|---------|
> | w.o load balancing   | 9.733%  | 34.34%  | 40.71%  |
> | w load balancing     | 9.104%  | 32.31%  | 38.56%  |
>
> |Personalized MoE| Hit@1   | Hit@5   | Hit@10  |
> |----------------------|---------|---------|---------|
> | w.o load balancing   | 9.733%  | 34.34%  | 40.71%  |
> | w load balancing     | 8.807%  | 31.57%  | 39.04%  |
>
> [1] Huang Q, An Z, Zhuang N, et al. Harder Task Needs More Experts: Dynamic Routing in MoE Models[C]//Proceedings of the 62nd Annual Meeting of the Association for Computational Linguistics (Volume 1: Long Papers). 2024: 12883-12895
>
> [2] Zoph B, Bello I, Kumar S, et al. St-moe: Designing stable and transferable sparse expert models[J]. arXiv preprint arXiv:2202.08906, 2022.
>
> [3] Fedus W, Zoph B, Shazeer N. Switch transformers: Scaling to trillion parameter models with simple and efficient sparsity[J]. Journal of Machine Learning Research, 2022, 23(120): 1-39.

---

> > ### Author Response · Authors · 2025-11-20
> > **Response to Reviewer Eynx (2/4)**
> >
> > > W2 The framework has a high dependence on a fixed set of five location functions and ten user groups. This raises concerns about its generalizability and scalability. It is unclear how well these pre-defined categories would transfer to new cities with different functional layouts (e.g., an industrial city vs. a tourist city).
> >
> > > Q2.Are these fixed categories still applicable when the model is applied to a new environment with a different urban layout (e.g., an industrial vs. a tourist city) or different cultural mobility patterns?
> >
> > We provide the following clarification regarding the generalizability and scalability of our predefined categories.
> >
> > First, we clarify that the location functions and user groups used in our framework are **not arbitrary choices**, but are based on **well-established and widely adopted** taxonomies in urban computing, mobility modeling, and urban functional zone analysis. The functional categories align with mainstream classifications in relevant researches [1–5]. The user groups strictly follow the design of [6], where the authors propose the same ten representative and high-coverage user groups to describe human mobility. Therefore, these categories represent **validated semantic abstractions rather than task-specific heuristics**, and possess **strong generality** across cities.
> >
> > Second, these predefined semantics **do not impose rigid or mutually exclusive structures** in our model. The MoE routing mechanism **activates multiple experts when necessary**, enabling each location or user to be represented as a compositional mixture of semantic directions rather than being forced into a single discrete category. This flexibility allows the model to **represent finer-grained or city-specific semantics as combinations of multiple experts**. For instance, an “industrial district” the reviewer mentioned might be naturally captured as a combination of multiple location-function experts. Similarly, mobility patterns of factory workers or night-shift employees can be expressed through suitable combinations of Office Worker, Night Shift Worker, and Service Industry Worker experts.
> >
> > Regarding tourist cities mentioned by the reviewer, their scenarios are also **covered** within our semantic space. Key components of tourism, such as attractions, cultural venues, commercial streets, dining areas, and hotels, correspond directly to Entertainment, Commercial, and Residential. Likewise, tourists are already modeled via the Visitor user group in Personalized MoE. Therefore, the existing functional categories and user groups provide complete coverage for tourism-driven urban environments without the need to introduce additional “tourism-specific” categories.
> >
> > From the **perspective of human mobility research**, even in strongly industrialized or strongly tourism-oriented cities, the vast majority of mobility patterns are still driven by stable and universal behavior semantics such as commuting, consumption, leisure, education, and public services. Only a fraction of trajectories correspond to “pure industrial production” or “pure tourism service” . Thus, our current semantic categories **naturally captures the dominant mobility patterns** across a wide range of city types.
> >
> > Currently, publicly available mobility datasets do not contain large-scale industrial cities, so we cannot empirically evaluate this scenario at present stage. If representative industrial-city trajectory data become available in the future, we will first assess whether the current semantic categories can adequately express such scenarios. If new industrial semantics cannot be sufficiently captured through combinations of existing experts, we can extend the expert sets in both Location Semantics MoE and Personalized MoE to explore the potential of introducing industrial-related location function or user-group experts.
> >
> > [1] Chen J, et al. Exploring Urban Functional Zones Based on Multi-source Semantic Knowledge and Cross-modal Network. The International Archives of the Photogrammetry, Remote Sensing and Spatial Information Sciences 2023.
> >
> > [2] Luo G,et al. Urban Functional Zone Classification Based on POI Data and Machine Learning. Sustainability 2023.
> >
> > [3] Hong Y, et al. Context-aware Multi-head Self-attentional Neural Network Model for Next Location Prediction. Transportation Research Part C: Emerging Technologies 2023.
> >
> > [4] Ma H, et al. Investigating Road-Constrained Spatial Distributions and Semantic Attractiveness for Area of Interest. Sustainability 2019.
> >
> > [5] Xiong Y, Li G. Correlation Characteristics Between Urban Fires and Urban Functional Spaces: A Study Based on POI Data and Ripley’s K-Function. ISPRS International Journal of Geo-Information 2025.
> >
> > [6] Wang J, Jiang R, Yang C, et al. Large language models as urban residents: An llm agent framework for personal mobility generation[J]. Advances in Neural Information Processing Systems, 2024, 37: 124547-124574.

---

> ### Author Response · Authors · 2025-11-20
> **Response to Reviewer Eynx (3/4)**
>
> > W3. Complex model. The system integrates multiple components (LLM, TCN, two MoEs, LoRA fine-tuning, and KD-Tree). It brings an extremely high cost, which may lead to problems such as difficulties in adjusting hyperparameters and reduced reproducibility.
>
> Although NextLocMoE conceptually contains multiple components, its overall design is **highly modular and easy to reproduce**. In terms of **training cost, inference efficiency, hyperparameter sensitivity, and component necessity**, the effective complexity of NextLocMoE is substantially lower than that of existing LLM-based methods.
>
> To begin with, NextLocMoE does **not** perform full-parameter fine-tuning on a 3B LLM. The backbone retains only the first 12 layers, and within these, only two FFNs in the top four layers are replaced by Personalized MoE. All remaining attention and FFN layers are **fully frozen**. During training, the only learnable parameters inside LLM are the LayerNorm weights and the low-rank LoRA parameters within Personalized MoE. Location Semantics MoE consists solely of lightweight linear mappings, and TCN used for historical encoding has negligible computational cost. The KD-tree is a deterministic post-processing step used only for inference. As detailed in Appendix U, **the total number of trainable parameters is 47.6M**, a very small fraction of the LLM backbone.
>
> Regarding actual **inference cost**, NextLocMoE is substantially more efficient than other LLM-based baselines. As reported in Table 3, the entire Kumamoto test set can be processed in 268 seconds using batch-parallel GPU inference, approximately 600× faster than Llama-Mob and 120× faster than LLMMob. This demonstrates that the conceptual “complexity” of NextLocMoE does not lead to computational burden; Moreover, Personalized MoE activates **fewer than two experts** on average across all cities (Fig.6(b)), making its actual inference cost close to that of a single FFN layer.
>
> For **hyperparameter tuning**, we performed extensive sensitivity analyses. Appendix P shows that key hyperparameters, such as the number of frozen layers and the historical length M, exhibit **stable performance over wide ranges**, indicating that NextLocMoE does not rely on delicate tuning and is robust to hyperparameter choices.
>
> It is also important to emphasize that the components of our system are **not arbitrary additions**, but are carefully designed to address two challenges in mobility prediction: multi-functional location semantics and heterogeneous user behaviors. Location Semantics MoE captures the multi-functionality of individual locations, while Personalized MoE models diverse mobility patterns across user groups. Ablation results confirm that removing either MoE leads to a clear performance drop, and removing the historical-aware router destabilizes expert assignment and further degrades performance.
>
> Finally, with respect to **reproducibility**, we have released the full codebase along with all configuration files. The appendix also provides details on hyperparameters, functional semantic priors, and user-group priors, ensuring that our results can be reproduced.
>
>
> > Q1: Could the authors provide a qualitative analysis of the expert activation frequencies? For example, showing the activation rates of the five location experts for a specific type of user trajectory.
>
> Thank you for the insightful suggestion. We have supplemented the expert activation patterns of Location Semantics MoE across different user groups on Singapore dataset. In Appendix S, we visualize the average activation distributions for four representative user groups (Student, Visitor, Office Worker, and Public Service Official), as well as for all users.
>
> The results reveal clear and semantically coherent activation differences across user groups. Student trajectories predominantly activate the Education expert; Visitors show a strong preference for the Entertainment expert; Office Workers choose the Commercial expert more frequently; and Public Service Officials activate the Public Service expert most prominently. These activation patterns demonstrate that NextLocMoE selects experts based on the mobility patterns reflected in user trajectories, leading to a clear and distinguishable division of semantic roles.

---

> ### Author Response · Authors · 2025-11-20
> **Response to Reviewer Eynx (4/4)**
>
> > Q3. How much does the quality of LLM semantic priors affect performance? Have you experimented with comparing the results (Detailed, rich descriptions vs. using simple labels (e.g., just "Commercial" or "Student")) for semantic priors? Or the influence of a slight perturbation of semantic priors on the results?
>
> Thank you for the valuable advice. We have added a controlled experiments (Appendix T) to evaluate how the quality of natural language priors affects model performance.
>
> We first replaced the detailed descriptions with extremely simple labels such as “This location belongs to Entertainment.” or “This user group is student.” The results show a **small and consistent decrease**, showing that richer priors provide clearer semantic guidance, but the model itself does **not** rely heavily on the exact description.
>
> We then examined the contribution of semantic components by removing specific parts of the descriptions (functional definitions (S1), typical examples (S2), or behavioral patterns (S3) for locations; identity cues (S1) or temporal & behavioral patterns (S2) for user groups). Removing any component leads to a **mild performance decline**, and the degradation becomes slightly larger as richer semantic information is removed. This indicates that each semantic component provides useful guidance, while the model remains **robust** and does not rely disproportionately on any single element.
>
> In summary, LLM-based semantic priors provide a **stable and positive performance gain** by giving both MoE modules clearer semantic information, but NextlLocMoE is not overly dependent on them. The effectiveness of NextLocMoE stems primarily from its dual-level MoE design and historical-aware routing mechanism.
>
> ### Simple semantics influence:
>
> |Location Semantics MoE| Hit@1   | Hit@5   | Hit@10  |
> | -------------- | ------- | ------- | ------- |
> | Rich label     | 9.733%  | 34.34%  | 40.71%  |
> | Simple label   | 9.511%   | 33.78%   | 40.42%   |
>
> |Personalized MoE| Hit@1   | Hit@5   | Hit@10  |
> | -------------- | ------- | ------- | ------- |
> | Rich label     | 9.733%  | 34.34%  | 40.71%  |
> | Simple label   | 9.347%   | 33.60%   | 40.25%   |
>
> ### Influence of semantic components:
>
> | Location semantics MoE| Hit@1   | Hit@5   | Hit@10  |
> | -------------- | ------- | ------- | ------- |
> | NextLocMoE     | 9.733%  | 34.34%  | 40.71%  |
> | w.o S1         | 9.709%   | 34.28%   | 40.66%   |
> | w.o S2         | 9.677%   | 34.23%   | 40.64%   |
> | w.o S3         | 9.638%   | 34.16%   | 40.58%   |
>
>
>
> |Personalized MoE| Hit@1   | Hit@5   | Hit@10  |
> | -------------- | ------- | ------- | ------- |
> | NextLocMoE     | 9.733%  | 34.34%  | 40.71%  |
> | w.o S1         | 9.611%   | 34.17%   | 40.59%   |
> | w.o S2         | 9.573%   | 34.09%   | 40.52%   |
>
> >Q4: Considering the overall complexity of the model, have you ever explored simplifying the model? For instance, if only one of the MoE modules is used, or if a simpler historical encoder (rather than a TCN) is used to boot the router, can a competitive balance be achieved between performance and complexity?
>
> Thank you for the thoughtful comments regarding model complexity. We have indeed conducted analysis on the potential simplification of our architecture, and the corresponding results are reported in Appendix J and Appendix N. These experiments already cover the two types of simplification suggested by the reviewer: reducing the number of MoE modules and replacing the historical encoder.
>
> Regarding the MoE structure, we examine models where either Location Semantics MoE or Personalized MoE is removed (Appendix J). The results show that removing **either** module leads to performance drop, indicating that the two MoE components encode **complementary and non-redundant** semantics. The Location Semantics MoE enhances the representation of multi-functional locations, while the Personalized MoE models heterogeneous user behavior patterns; neither can substitute for the other.
>
> For the historical encoder, we evaluated several simpler alternatives, including completely removing historical encoding and replacing TCN with LSTM or a lightweight attention-based encoder (Appendix N). All three alternatives result in **degradation** in both accuracy and transfer performance. This demonstrates that TCN is the most suitable historical encoder in NextLocMoE, largely due to its ability to efficiently capture temporal patterns in mobility trajectories.

---

> ### Author Response · Authors · 2025-11-27
> **Gentle Reminder**
>
> Dear Reviewer Eynx,
>
> We sincerely thank you again for your valuable comments and constructive suggestions. As discussion stage is approaching its end, we would appreciate it if you could please let us know whether our responses and the revised manuscript have addressed your concerns, and let us know if any issues remain.
>
> Best regards,
>
> Author of Submission 86

---

### Official Review · Reviewer_93Nk · 2025-10-29

**Soundness:** 2
**Presentation:** 4
**Contribution:** 3
**Rating:** 4
**Confidence:** 4

**Summary:**

This paper proposes an innovative framework for next-location prediction, NextLocMoE, which integrates large language models (LLMs) with a dual-layer Mixture-of-Experts (MoE) architecture to jointly model the multi-functional semantics of locations and the behavioral diversity of users. The framework consists of two key modules: Location Semantics MoE, which models multi-functional location semantics, and Personalized MoE, which captures individual behavioral preferences. In addition, a historical-aware router is introduced to enhance the stability of expert selection by incorporating temporal dependencies.

**Strengths:**

1. The paper introduces a well-structured and clearly motivated dual-layer MoE architecture that effectively captures both the multi-functional semantics of locations and the diversity of user behaviors. The architectural design is novel and well-executed.

2. The historical-aware routing mechanism is innovative and well-motivated. By integrating trajectory history into expert selection, it effectively models long-term behavioral dependencies and improves the stability of routing decisions.

**Weaknesses:**

1. It remains unclear how experts in NextLocMoE learn meaningful semantic roles or capture distinct user groups, since the model is trained without explicit supervision (e.g., no annotations for location functions or user groups) or loss terms that guide such specialization. Although the paper describes the existence of these roles, it does not explain how the experts are selected or differentiated.

2. The experimental setup for zero-shot prediction is not clearly described—specifically, the training and test dataset splits for baselines such as LLM-Mob and ZS-NL (in Table 2) are insufficiently detailed.

3. The cross-city generalization mechanism is not fully explained. It remains unclear how coordinate-based predictions can preserve region-level semantic information across datasets, especially when the same coordinates may have different meanings in different cities. The paper should further clarify how data and knowledge transfer are achieved in such scenarios.

**Questions:**

Please refer to weaknesses

---

> ### Author Response · Authors · 2025-11-20
> **Response to Reviewer 93Nk (1/3)**
>
> > W1: It remains unclear how experts in NextLocMoE learn meaningful semantic roles or capture distinct user groups, since the model is trained without explicit supervision (e.g., no annotations for location functions or user groups) or loss terms that guide such specialization. Although the paper describes the existence of these roles, it does not explain how the experts are selected or differentiated.
>
> Thank you for raising this question. We would like to clarify that, although NextLocMoE does not rely on annotated location-function labels or user-group labels, its experts are **not** trained in a purely unguided manner. Instead, NextLocMoE incorporates **semantic initialization, semantic routing inputs, historical-aware routing, and entropy regularization**, which together form a strong and stable **soft semantic prior** that naturally leads to differentiated and interpretable expert specializations.
>
> In Location Semantics MoE, the five function experts are **not randomly initialized**. Each expert is initialized using LLM-encoded embeddings derived from natural-language descriptions of representative functional categories. This semantic initialization provides a **clear semantic direction** for each expert. During training, its router tends to select experts whose semantic priors align with the historical trajectory patterns, in order to reduce coordinate prediction error. As a result, the location function experts naturally converge toward distinct functional roles rather than forming arbitrary clusters. Prior works have shown that such semantic initialization leads to **stable long-term specialization** and effectively prevent expert collapse [1][2][3][4].
>
> For Personalized MoE, each expert is associated with an LLM-encoded **user-group semantic prior**, which is provided as an additional input to the router. This guides the router to specialize experts toward distinct behavioral patterns. During training, the router learns which semantic direction best matches the user’s mobility pattern so as to minimize the prediction loss.
>
> Moreover, the historical-aware router in both MoE modules integrates long-term mobility signals into the routing process, ensuring that expert selection does not rely solely on short-term context. This significantly improves **routing stability**. In addition, in Personalized MoE, an entropy regularization term encourages **high-confidence decisions** rather than activating many experts simultaneously, further sharpening expert differentiation and enhancing interpretability.
>
> We hope this clarification resolves your concerns.
>
> [1] Gritsch N, Zhang Q, Locatelli A, et al. Nexus: Specialization meets adaptability for efficiently training mixture of experts[J]. arXiv preprint arXiv:2408.15901, 2024.
>
> [2] Zheng C, Huang M, Pedchenko D, et al. Enhancing embedding representation stability in recommendation systems with semantic id[C]//Proceedings of the Nineteenth ACM Conference on Recommender Systems. 2025: 954-957.
>
> [3] Park J, Ahn Y J, Kim K E, et al. Monet: Mixture of monosemantic experts for transformers[J]. arXiv preprint arXiv:2412.04139, 2024.
>
> [4] Qi Y, Sachan D, Felix M, et al. When and why are pre-trained word embeddings useful for neural machine translation?[C]//Proceedings of the 2018 Conference of the North American Chapter of the Association for Computational Linguistics: Human Language Technologies, Volume 2 (Short Papers). 2018: 529-535.

---

> > ### Author Response · Authors · 2025-11-20
> > **Response to Reviewer 93Nk (2/3)**
> >
> > > W2: The experimental setup for zero-shot prediction is not clearly described—specifically, the training and test dataset splits for baselines such as LLM-Mob and ZS-NL (in Table 2) are insufficiently detailed.
> >
> > Thank you for pointing out the need for a clearer description of zero-shot experimental setting. We have added more detailed explanations in our revised version (Appendix H), and provide a clarification here.
> >
> > All datasets used in our experiments follow a user-level partitioning scheme. Following [1], users are split into training, validation, and test sets with a ratio of 7:1:2. All baselines use the same split, ensuring comparison fairness.
> >
> > For cross-city models that require training (NextLocMoE, Llama-Mob, and NextLocLLM), we adopt a unified zero-shot setting: the model is trained **only** on the training set of the source city, and once training is completed, it is evaluated **directly** on the test set of the target city. **No training or validation data from the target city are used, and no fine-tuning or adaptation is performed at any stage**. For example, in the Shanghai→Kumamoto setting, all models are trained merely on the Shanghai training set and then evaluated on the Kumamoto test set.
> >
> > For prompt-based methods that do not require training (ZS-NL and LLMMob), we strictly follow their **original formulations**: since these methods do not involve a training phase, we directly run inference on the target city’s test set using their corresponding prompt templates.
> >
> > [1] Sun H, Xu J, Zheng K, et al. MFNP: A meta-optimized model for few-shot next POI recommendation[C]//IJCAI. 2021, 2021: 3017-3023.

---

> ### Author Response · Authors · 2025-11-20
> **Response to Reviewer 93Nk (3/3)**
>
> > W3: The cross-city generalization mechanism is not fully explained. It remains unclear how coordinate-based predictions can preserve region-level semantic information across datasets, especially when the same coordinates may have different meanings in different cities. The paper should further clarify how data and knowledge transfer are achieved in such scenarios.
>
> Thank you for raising this important point regarding cross-city generalization. Below, we provide a systematic clarification of how NextLocMoE maintains location-level semantic consistency across different cities, and we have added a detailed explanation in our revised manuscript (Appendix E).
>
> First, we clarify that our model **does not treat coordinates themselves as the carriers of location semantics**. Normalized coordinates simply place different cities into a comparable numerical range so that the location function experts can process them under a unified input scale. **Location semantics are not determined by coordinate values, but by Location Semantics MoE**, whose five function experts correspond to five representative location semantic categories. Each category has its natural-language description, which is encoded using LLM to initialize the parameters of its corresponding expert. These experts are shared across all cities, providing a **consistent semantic basis** that does not depend on any particular city’s coordinate system.
>
> Second, NextLocMoE infers semantics through a **historical-aware routing mechanism** rather than coordinate mapping. The router takes as input both the initial embedding of current location and the user’s historical behavioral representation, and learns which function experts should be activated. Thus, **the same coordinates can lead to different expert activation and different location semantics because the historical trajectory patterns differ**.
>
>
> Under this mechanism, cross-city transfer does **not** require coordinate alignment. During training on the source city, the router learns mappings from historical trajectory patterns to function-expert activation. When transferred to target city, the location function experts remain semantically consistent, while the learned router **allocates expert activation weights based on the given historical trajectories**, enabling reasonable semantic interpretations without requiring labels from the new city.
>
> Finally, our choice of normalized coordinates rather than raw coordinates or location IDs follows naturally from this mechanism. Raw coordinates differ greatly in scale and range across cities, making it difficult for shared experts to learn consistent spatial mappings; location IDs are city-specific and cannot generalize to unseen cities. Normalized coordinates provide a unified geometric scale, allowing the shared experts to apply consistent mappings across cities, while **location semantics are dynamically activated by the MoE router** based on mobility behavioral patterns. Therefore, **the key to cross-city transfer lies not in coordinate alignment, but in the cross-city consistent semantics encoded by location function experts and the historical-aware router’s adaptive expert activation.**

---

> > ### Author Response · Authors · 2025-11-27
> > **Gentle Reminder**
> >
> > Dear Reviewer 93Nk,
> >
> > We sincerely thank you again for your valuable comments and constructive suggestions. As discussion stage is approaching its end, we would appreciate it if you could please let us know whether our responses and the revised manuscript have addressed your concerns, and let us know if any issues remain.
> >
> > Best regards,
> >
> > Author of Submission 86

---

### Author Response · Authors · 2025-11-20
**General response to all reviewers**

We sincerely thank all reviewers for their thoughtful and constructive feedback. We are pleased that our dual-MoE architecture is recognized as **well-structured, clearly motivated, and novel** (Reviewers 93Nk, Eynx, WfzH). We also appreciate the positive assessments of our historical-aware routing mechanism, which reviewers found **innovative and well-justified** (Reviewer 93Nk). We are glad that multiple reviewers highlighted the **interpretability** of our experts (Reviewers Eynx, WfzH), as well as the **robustness of our experimental results** (Reviewers Eynx, hP3j, WfzH). We further thank Reviewer hP3j for acknowledging the **clarity** of our writing and presentation.

Following the reviewers’ valuable suggestions, we have substantially revised our manuscript and added several new experiments and analyses. All changes are highlighted in blue in the revised version for ease of inspection.

Key updates include:

+ A systematic clarification of how NextLocMoE preserves location-level semantic consistency across different cities (Appendix E).

+ A more precise and detailed explanation of our zero-shot experimental setting (Appendix H).

+ Hyperparameter sensitivity analysis of the historical length M (Appendix P).

+ Comparison of models with vs. without load-balancing, including predictive performance and expert-activation analyses (Appendix R).

+ Visualizations of Location Semantics MoE activations for representative user groups (Appendix S).

+ Evaluation of how the quality of natural-language semantic priors affects model performance (Appendix T).

+ Calculation of the full number of trainable parameters (Appendix U).

We hope that these responses and revisions adequately address all reviewer concerns and further strengthen the contribution of our work.

---

### Meta-Review · Area_Chair_aVvm · 2025-12-22

**Summary:**

This paper proposes NextLocMoE for next-location prediction. The core idea is a two-level MoE architecture to incorporate both semantics and personalization: semantic experts are used to model location semantics, while personalization experts are used to model user differences. The paper further claims that soft priors can encourage different experts to acquire differentiated semantic/personalized knowledge. The authors also emphasize advantages in both zero-shot performance and inference efficiency, and provide a relatively comprehensive set of experiments and analyses.

The reviews exhibit substantial disagreement. Some reviewers appreciate the problem setting and the engineering effectiveness and give clearly positive evaluations, while others argue that the core mechanism lacks verifiability and irreplaceability, particularly whether the semantic/personalization MoE truly learns generalizable expert knowledge, and the extent to which the implementation relies heavily on a predefined category system. The rebuttal strengthens parts of the experiments and clarifies some descriptions, but the key disputes around the core mechanism are still not sufficiently closed with direct evidence.

Therefore, I lean toward a rejection.

**Reviewer Concerns:**

**A. Points that are partially addressed/mitigated in the rebuttal**

1. **Additional experiments and clarifications**: The authors further explain implementation details, efficiency analyses, and experimental settings, which helps readers understand the engineering motivation and the intended applicability.
2. **Some added/strengthened ablations and comparisons**: The rebuttal adds or reinforces certain comparisons and ablations, making the overall effectiveness of the model clearer.

These additions improve the completeness of the work, but they primarily enhance presentation and coverage, and provide limited help in resolving the core-mechanism concerns below.

**B. Remaining concerns**

1. **Dependence on predefined semantic/personalization categories**
   Multiple reviewers point out that the semantic and personalization MoE design relies heavily on a predefined category system (e.g., a predefined partition of location functions/semantics and predefined groupings of user populations), and uses it to construct soft priors that guide routing or expert learning. This further leads to the following two core disputes.
2. **The claim that “soft priors encourage different experts to learn differentiated semantic/personalized knowledge” lacks a verifiable, closed-loop validation**
   Reviewers question whether such indirect supervision can indeed yield distinct semantic knowledge across experts. While the authors claim that soft priors and routing can guide expert knowledge formation, this currently reads more like a plausible intuition than a hard, mechanism-level validation. For example:
   - Do experts truly exhibit stable specialization, and can this be demonstrated with quantitative metrics?
   - If the prior categories are removed or randomized, can experts still self-organize into an interpretable semantic decomposition (or does performance drop significantly, thereby showing the prior is a key dependency)?
   - Does the same expert maintain a semantically consistent role across different cities and different data splits?
3. **The generalization boundary of personalization modeling remains unclear**
   Since the expert knowledge depends on a predefined category system, the boundary of cross-city generalization and portability should be specified more clearly. At present, there is insufficient systematic evidence to support the robustness of “personalization experts” in open environments. The authors mention in the rebuttal that more complex/unknown scenarios could be handled by composing predefined groupings, but systematic experiments are still lacking to delineate: (i) the effective boundary of such composition and transfer; and (ii) whether experts can remain stably specialized and interpretable as the complexity of composition and routing increases.

Across the review, the central dispute is whether the observed improvements can be attributed to a generalizable mechanism contribution. The current evidence chain is still insufficient to directly support the paper’s key claims about the semantic/personalization MoE mechanism. In fact, many rebuttal experiments touch this question only indirectly. For instance, the activation patterns across different groups in Appendix S provide some support for expert differentiation; however, the strong impact of the w/o load-balancing regularization discussed in Appendix R also suggests that expert specialization may depend heavily on training regularization and optimization details, rather than being driven primarily by the claimed soft priors. Taken together, these indicate that the mechanism attribution for “soft-prior-driven expert semantic formation” still lacks direct, exclusive validation, and readers are required to infer the attribution on their own.

**Reviewer Scores:**

The scores are polarized. Notably, reviewers with positive conclusions did not further provide more detailed arguments or additional supporting evidence on the key disputed points, while the core concerns raised by low-score reviewers may be partially mitigated by the rebuttal but have not been directly resolved.

---

### Decision · Program_Chairs · 2026-01-26

Reject